# The mechanism of MICU-dependent gating of the mitochondrial Ca$^{2+}$ uniporter

**Vivek Garg[1,2]\*, Junji Suzuki[1], Ishan Paranjpe[1], Tiffany Unsulangi[1], Liron Boyman[2], Lorin S Milescu[3], W Jonathan Lederer[2], Yuriy Kirichok[1]\***

[1]Department of Physiology, University of California San Francisco, San Francisco, United States; [2]Department of Physiology, University of Maryland, Baltimore, United States; [3]Department of Biology, University of Maryland, College Park, United States

**Abstract** Ca$^{2+}$ entry into mitochondria is through the mitochondrial calcium uniporter complex (MCU$_{cx}$), a Ca$^{2+}$-selective channel composed of five subunit types. Two MCU$_{cx}$ subunits (MCU and EMRE) span the inner mitochondrial membrane, while three Ca$^{2+}$-regulatory subunits (MICU1, MICU2, and MICU3) reside in the intermembrane space. Here, we provide rigorous analysis of Ca$^{2+}$ and Na$^{+}$ fluxes via MCU$_{cx}$ in intact isolated mitochondria to understand the function of MICU subunits. We also perform direct patch clamp recordings of macroscopic and single MCU$_{cx}$ currents to gain further mechanistic insights. This comprehensive analysis shows that the MCU$_{cx}$ pore, composed of the EMRE and MCU subunits, is not occluded nor plugged by MICUs during the absence or presence of extramitochondrial Ca$^{2+}$ as has been widely reported. Instead, MICUs potentiate activity of MCU$_{cx}$ as extramitochondrial Ca$^{2+}$ is elevated. MICUs achieve this by modifying the gating properties of MCU$_{cx}$ allowing it to spend more time in the open state.

**\*For correspondence:**
vgarg@som.umaryland.edu (VG);
yuriy.kirichok@ucsf.edu (YK)

**Competing interests:** The authors declare that no competing interests exist.

## Introduction

Mitochondrial Ca$^{2+}$ uptake regulates ATP production by modulating the activities of several dehydrogenases in the mitochondrial matrix primarily the pyruvate dehydrogenase and likely other control systems (*Glancy and Balaban, 2012*; *McCormack et al., 1990*; *McCormack and Denton, 1993*; *Wescott et al., 2019*). Matrix Ca$^{2+}$ ([Ca$^{2+}$]$_m$) also plays a crucial role in influencing cell fate (*Bernardi, 1999*; *Berridge et al., 2003*; *Glancy and Balaban, 2012*; *Gunter et al., 2010*). Physiological and pathological Ca$^{2+}$ signaling in mitochondria depend on Ca$^{2+}$ entry into the matrix (*Holmström et al., 2015*; *Kwong et al., 2015*; *Luongo et al., 2017*) and its extrusion through the mitochondrial sodium-calcium exchanger (*Boyman et al., 2013*; *Luongo et al., 2015*; *Palty et al., 2010*) and other mechanisms (*Bernardi, 1999*; *Gunter et al., 2010*). Ca$^{2+}$ entry is mediated by the mitochondrial Ca$^{2+}$ uniporter holocomplex (MCU$_{cx}$) (*Bernardi, 1999*; *Deluca and Engstrom, 1961*; *Gunter et al., 2010*), a Ca$^{2+}$-selective channel that is regulated by the intracellular (extra-mitochondrial) [Ca$^{2+}$] level ([Ca$^{2+}$]$_i$) (*Fieni et al., 2012*; *Kirichok et al., 2004*). The MCU$_{cx}$ is composed of five distinct subunits types, two of which span the inner mitochondrial membrane (IMM) - MCU and EMRE - and two of the three MICU subunits (MICU1, MICU2, and MICU3) which reside in the intermembrane space (*Baughman et al., 2011*; *De Stefani et al., 2011*; *Sancak et al., 2013*). MICU1 connects an EMRE subunit in the MCU$_{cx}$ with a second MICU subunit. Recent structural discoveries (*Wang et al., 2019*; *Fan et al., 2020*; *Wang et al., 2020b*; *Zhuo et al., 2021*) suggest that the functioning channel is a dimer composed of two MCU/EMRE pores joined through the N-terminal of MCU subunits in the matrix, and MICU subunits in the intermembrane space.

For Ca$^{2+}$ to enter the matrix, Ca$^{2+}$ must first permeate the outer mitochondrial membrane (OMM) through the largely open VDAC (voltage-dependent 'anion' channel), a beta-barrel channel into the intermembrane space (IMS). From the IMS, Ca$^{2+}$ crosses the nearly impermeant inner

mitochondrial membrane (IMM) in a highly regulated manner into the mitochondrial matrix through the small conductance, highly selective MCU$_{cx}$ (*Kirichok et al., 2004*; *Williams et al., 2013*). The recent dynamic and exciting body of work investigating Ca$^{2+}$ movement through the MCU$_{cx}$ has led to a number of controversial and perplexing reports (*Csordás et al., 2013*; *Tomar et al., 2019*; *Foskett and Madesh, 2014*; *Gottschalk et al., 2019*; *Hoffman et al., 2013*; *Mallilankaraman et al., 2012b*; *Nemani et al., 2018*; *Perocchi et al., 2010*; *Tufi et al., 2019*; *Supplementary file 1a and b*). These publications also provoke the possibility that the molecular components of MCU$_{cx}$ have additional broad actions in mitochondria which could complicate our understanding (*Tomar et al., 2019*; *Gottschalk et al., 2019*; *Tufi et al., 2019*). Here, we use an array of quantitative tools to directly examine the conductance of the MCU$_{cx}$ channel and how it is gated by MICU subunits. Our investigation provides reasons to question some of the published working hypotheses and suggest a new view of the molecular gating of MCU$_{cx}$.

## Results

### Quantitative assessment of the MCU$_{cx}$ and its subunits

A whole mitoplast patch clamp method was used to measure whole IMM current to assess MCU$_{cx}$ function (*Fieni et al., 2012*; *Garg and Kirichok, 2019*; *Kirichok et al., 2004*) and determine how the subunits contribute to the measured MCU$_{cx}$ current. Mitochondria were isolated from DRP1 knock-out (KO) mouse embryonic fibroblasts (MEFs) (*Ishihara et al., 2009*). DRP1 is encoded by the *Dnm1l*. The DRP1-KO MEFs were used to prepare mitoplasts using a French Press. This cell line was chosen as the source for many experiments because it provided a significantly higher proportion of large isolated mitoplasts and enabled the recording of stable MCU$_{cx}$ currents with a favorable signal-to-noise ratio. We confirmed that this cell line expresses all principal subunits of the MCU complex (*Figure 1—figure supplement 1*). Importantly, the MCU$_{cx}$ was intact in isolated mitoplasts, and its composition was the same as in intact mitochondria (*Figure 1—figure supplement 2F*). We also generated gene knockouts for all principal subunits of the MCU complex (MCU, EMRE, and MICU1−3) using CRISPR-Cas9 in this cell line (*Figure 1—figure supplement 1*). MCU, EMRE, and MICU1−3 are encoded by the *Mcu*, *Smdt1* and *Micu1−3* genes, respectively.

*Figure 1A* shows the [Ca$^{2+}$]$_i$ dependence of the MCU$_{cx}$ current in mitoplasts from WT DRP1-KO MEFs, and shows the absence of Ca$^{2+}$ current ($I_{Ca}$) in MCU-KO or EMRE-KO. Additionally, it shows an important feature of the MCU$_{cx}$; in the absence of extramitochondrial Ca$^{2+}$ (control trace), there is outward current at positive potentials resulting from the efflux of Na$^+$ through the MCU$_{cx}$ due to the 110 mM Na$^+$ gluconate in the matrix from the patch pipette (*Figure 1A*, *Figure 1—figure supplement 2D and E*). When extramitochondrial Ca$^{2+}$ is present, Ca$^{2+}$ enters the selectivity filter of the MCU$_{cx}$ channel to block Na$^+$ permeation (*Fieni et al., 2012*; *Garg and Kirichok, 2019*; *Kirichok et al., 2004*) and no outward current is seen. Importantly, MCU$_{cx}$ currents can be rescued by the ectopic expression of the MCU and EMRE subunits in their corresponding knockout cell lines (*Figure 1B and C*, *Figure 1—figure supplement 2G and H*). From these results, the DRP1-KO MEFs recapitulate key findings in previous publications (*Chaudhuri et al., 2013*; *Fieni et al., 2012*; *Kirichok et al., 2004*; *Sancak et al., 2013*). We show an additional novel observation, important to our later experiments, that when Na$^+$ is used to replace Ca$^{2+}$ in the cytosolic compartment (i.e. the bath solution), an MCU-mediated Na$^+$ current ($I_{Na}$) is observed, and this current also depends on the presence of MCU and EMRE (*Figure 1D and E*, *Figure 1—figure supplement 2D and E*).

In intact WT cells, the [Ca$^{2+}$]$_i$ increase (elicited by SERCA inhibitor thapsigargin) was followed, after a short delay, by [Ca$^{2+}$]$_m$ elevation as detected by a genetically-encoded Ca$^{2+}$ indicator *Cepia* targeted to mitochondria (*Suzuki et al., 2014*; *Figure 1—figure supplement 2A*). However, as expected, in MCU-KO or EMRE-KO cell lines that have no functional MCU$_{cx}$ (*Baughman et al., 2011*; *De Stefani et al., 2011*; *Sancak et al., 2013*), no significant [Ca$^{2+}$]$_m$ elevation was observed (*Figure 1—figure supplement 2B and C*).

One of the controversial elements in previously published experiments is the explanation of the cause of the 'threshold' of the MCU$_{cx}$ Ca$^{2+}$ influx into the matrix (*Csordás et al., 2013*; *Tomar et al., 2019*; *Foskett and Madesh, 2014*; *Hoffman et al., 2013*; *Mallilankaraman et al., 2012b*; *Perocchi et al., 2010*; *Tufi et al., 2019*). It was noted initially (*Mallilankaraman et al., 2012b*) that there is a cytosolic concentration of Ca$^{2+}$ ([Ca$^{2+}$]$_i$) below which there is no MCU$_{cx}$-

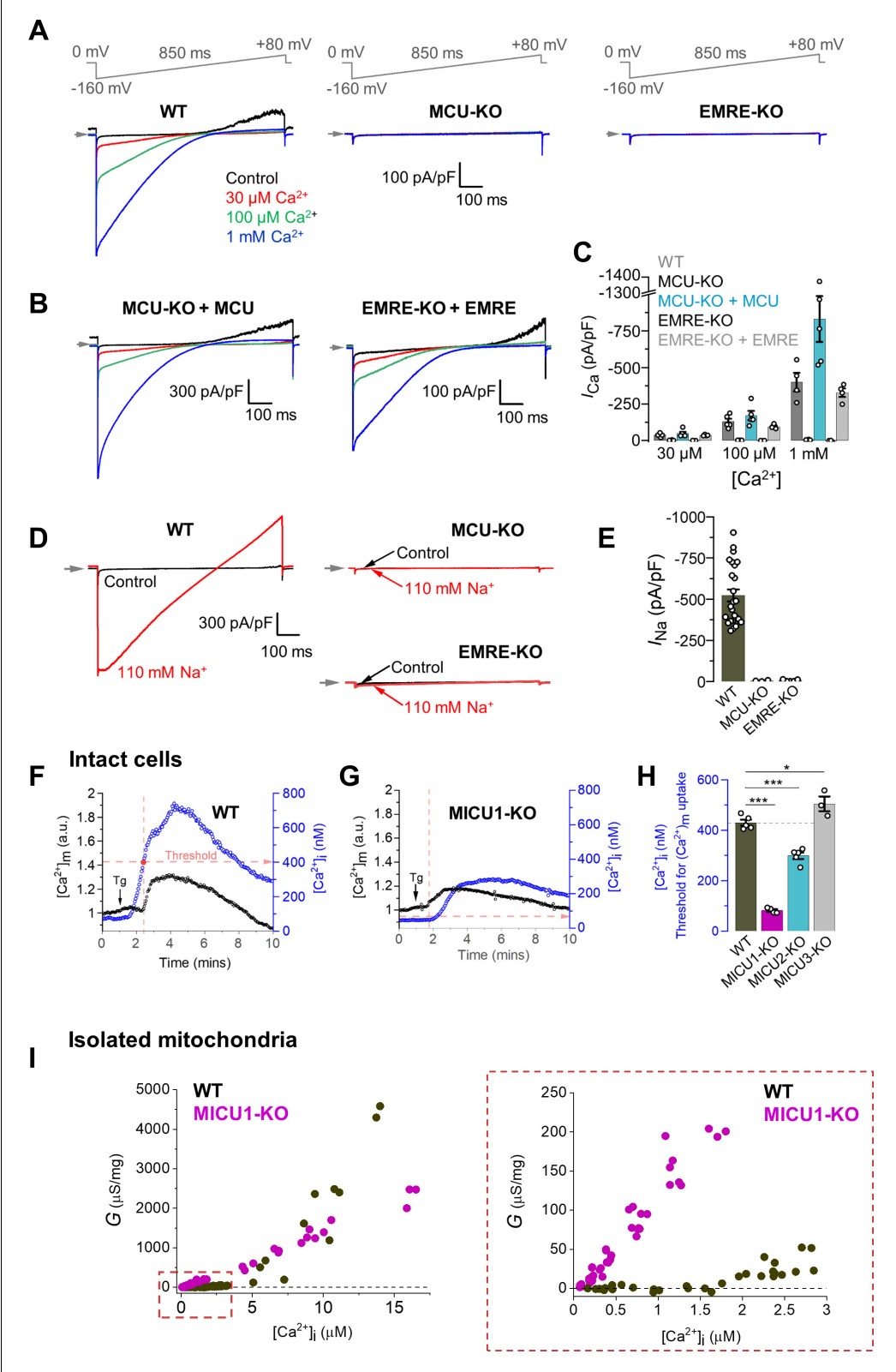

**Figure 1.** Characterization of MCU$_{cx}$ properties in isolated mitoplasts, intact cells, and isolated mitochondria from MEFs. (**A**) Inward $I_{Ca}$ elicited by a voltage ramp in WT, MCU-KO and EMRE-KO mitoplasts exposed to [Ca$^{2+}$]$_i$ of 30 µM, 100 µM, and 1 mM. In WT, also note an outward Na$^+$ current via MCU at positive voltages in Ca$^{2+}$-free bath solution (Control). Voltage protocol is indicated on the top. All superimposed current traces in a single panel are from the same mitoplast. (**B**) $I_{Ca}$ is rescued by the recombinant expression of MCU and EMRE in their respective knockout cell lines. (**C**) $I_{Ca}$

*Figure 1 continued on next page*

*Figure 1 continued*

density measured at −160 mV at different $[Ca^{2+}]_i$ in indicated cell lines; n = 4–5. (**D**) Representative $I_{Na}$ in WT, MCU-KO and EMRE-KO mitoplasts at 110 mM $[Na^+]_i$. (**E**) $I_{Na}$ density measured at −80 mV in WT, MCU-KO, and EMRE-KO mitoplasts; n = 3–20. (**F and G**) Representative $[Ca^{2+}]_m$ (*black*, left ordinate) and $[Ca^{2+}]_i$ (*blue*, right ordinate) in an individual cell with (**F**) WT MCU$_{cx}$, and (**G**) MICU1 knockout before and after application of 300 nM thapsigargin (Tg, arrow). Dashed red lines indicate the $[Ca^{2+}]_i$ at which the $[Ca^{2+}]_m$ starts to increase ('Threshold'). (**H**) $[Ca^{2+}]_i$ threshold for $[Ca^{2+}]_m$ elevation in WT and indicated knockout cell lines; n = 3–4 dishes, total cells = ~150 each group. Data shown as mean ± SEM; one-way ANOVA with post-hoc Tukey test. Statistics was run on number of dishes. (**I**) $Ca^{2+}$ conductance (G) of the IMM plotted as a function of $[Ca^{2+}]_i$. Right panel shows the zoomed-in region for $[Ca^{2+}]_i$ between 0 and 3 µM; n = 64–75 independent experiments, N = 4–7 independent preparations, all data is shown. All superimposed current traces in a single panel are from the same mitoplast. Data shown as mean ± SEM.

The online version of this article includes the following source data and figure supplement(s) for figure 1:

**Source data 1.** Dataset values for *Figure 1*.
**Figure supplement 1.** Generation of knockouts for various MCU$_{cx}$ subunits.
**Figure supplement 1—source data 1.** Raw western blot image for panel G.
**Figure supplement 1—source data 2.** Raw western blot image for panel H.
**Figure supplement 1—source data 3.** Raw western blot image for panel I.
**Figure supplement 1—source data 4.** Real time PCR (Ct) values for different genes.
**Figure supplement 2.** $[Ca^{2+}]_m$ phenotype in cells deficient for various MCU$_{cx}$ subunits, patch clamp methodology, and protein expression of various MCU$_{cx}$ subunits in isolated mitoplasts and MEFs.
**Figure supplement 2—source data 1.** Raw western blot image for panel F.
**Figure supplement 2—source data 2.** Raw western blot image for panel G.
**Figure supplement 2—source data 3.** Raw western blot image for panel H.
**Figure supplement 3.** Mitochondrial $Ca^{2+}$ uptake phenotype in cells and mitochondria deficient for MICU subunits.
**Figure supplement 3—source data 1.** Dataset values for *Figure 1—figure supplement 3*.

medicated $Ca^{2+}$ influx. *Figure 1F* shows that such a threshold for DRP1-KO WT MEF is indeed found at around 400 nM $Ca^{2+}$. This threshold is largely gone in MICU1-KO cells (*Figure 1G*). In MICU2-KO and MICU3-KO cells, the changes in the threshold levels are shown in *Figure 1H*, and *Figure 1—figure supplement 3A–D*. Past studies of other investigators have used similar information from their MICU1-KO cells to argue that MICU1 forms a 'plug' or an occlusion in the channel pore. This implies that in MICU1-KO cells, the MCU$_{cx}$ conductance should be greater due to the removal of the plug. This conclusion, however, is challenged by the $Ca^{2+}$ conductance studies shown here in isolated mitochondria from WT MEFs shown in *Figure 1I* and *Figure 1—figure supplement 3E*. While the Figure shows increased conductance of MCU$_{cx}$ at low $[Ca^{2+}]_i$ in the MICU1-KO mitochondria consistent with the removal of a putative MCU$_{cx}$'plug', there is decreased conductance at high $[Ca^{2+}]_i$ (~8 µM or higher), an observation that is inconsistent with the plug hypothesis (*Mallilankaraman et al., 2012b*). Additional recent modifications of this hypothesis add the prediction that allosteric actions of MICU1 on MCU$_{cx}$ account for any inconsistencies or contradictions of the plug hypothesis (*Csordás et al., 2013*). Moreover, new findings suggest that there may be broad actions of MICU1 on non-MCU$_{cx}$ targets within the mitochondria (*Tomar et al., 2019*; *Gottschalk et al., 2019*; *Tufi et al., 2019*). These findings and the unrefined and untested modifications of the plug hypothesis motivate additional investigations. We have carried out new quantitative experiments and analysis that may help us better understand how MICU1 works in the context of the MCU$_{cx}$ as is shown in *Figure 2*.

## MICUs are $[Ca^{2+}]_i$-dependent MCU$_{cx}$ potentiators

To investigate how MICU1 works, $I_{Ca}$ was measured in mitoplasts at five extramitochondrial $[Ca^{2+}]_i$ levels, 10 µM, 100 µM, 1 mM, 5 mM, and 25 mM (*Figure 2A and B*, and *Figure 2—figure supplement 1A–C*). WT mitoplasts show $I_{Ca}$ records similar to the current density measurements from MICU2-KO and MICU3-KO mitoplasts. In contrast, WT mitoplasts have $I_{Ca}$ current densities that are roughly twice the size of the current densities from the MICU1-KO mitoplasts. This finding is like the conductance measurements at elevated $[Ca^{2+}]_i$ in *Figure 1I* and thus inconsistent with the plug hypothesis that posits that MICU1 is an obstructing plug of MCU$_{cx}$.

The expression of EMRE protein (but not MCU) was significantly reduced in MICU1-KO (*Figure 2—figure supplement 1D–F*), as was also shown previously (*Liu et al., 2016*). However, the lower EMRE expression in MICU1-KO was not a limiting factor for $I_{Ca}$, because EMRE overexpression

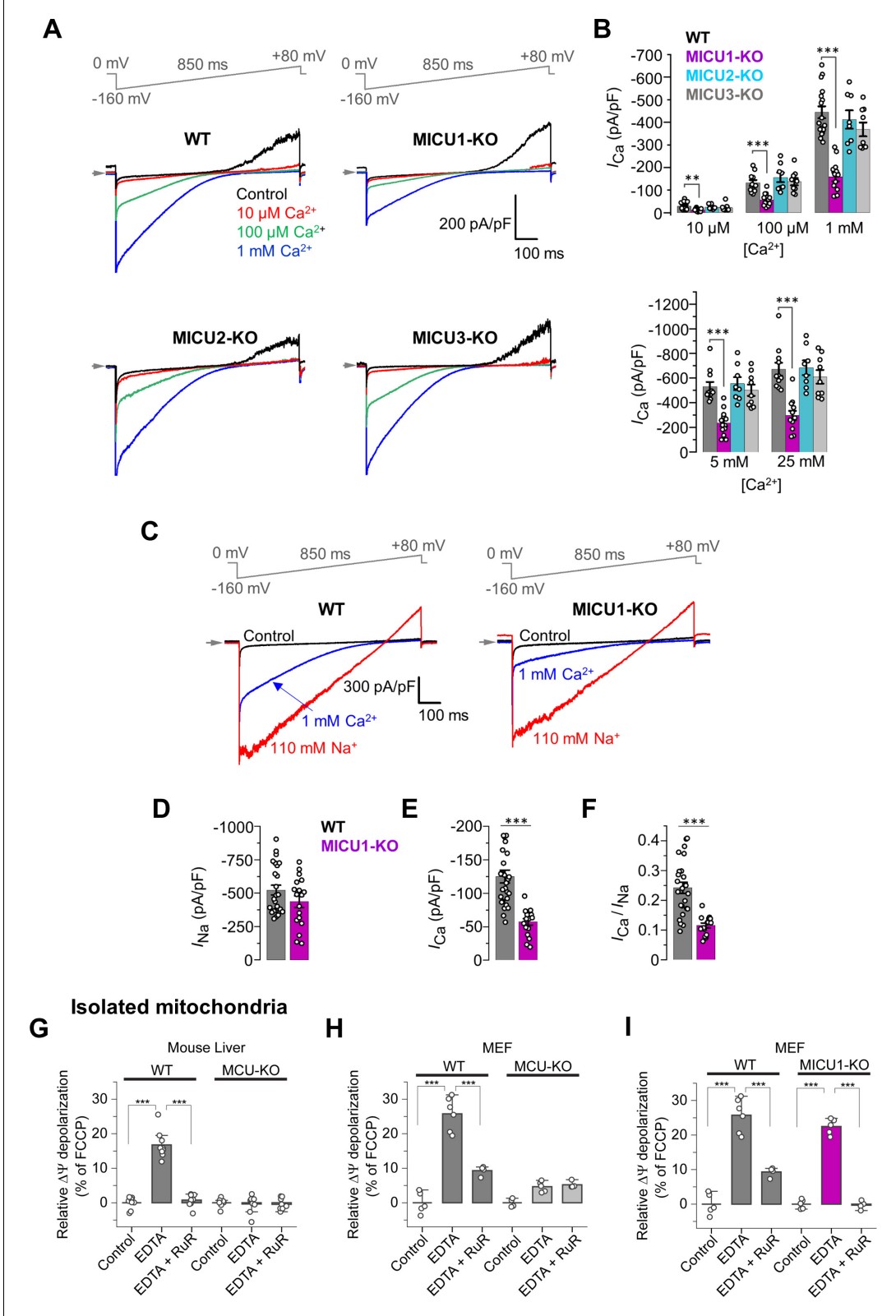

**Figure 2.** MICUs are $[Ca^{2+}]_i$-dependent $MCU_{cx}$ potentiators. (**A**) Inward $I_{Ca}$ in WT, MICU1-KO, MICU2-KO, and MICU3-KO mitoplasts exposed to 10 μM, 100 μM and 1 mM $[Ca^{2+}]_i$. (**B**) $I_{Ca}$ densities measured at −160 mV and $[Ca^{2+}]_i$ of 10 μM, 100 μM, and 1 mM (*upper*), as well as 5 mM and 25 mM (*lower*). Data shown as mean ± SEM; one-way ANOVA with post-hoc Tukey test, n = 8–17. (**C**) Representative $I_{Ca}$ (*blue*) and $I_{Na}$ (*red*) recorded from the same WT and MICU1-KO mitoplasts exposed to 1 mM $[Ca^{2+}]_i$ or 110 mM $[Na^+]_i$ in the absence of $Ca^{2+}$. (**D–F**) Amplitudes of $I_{Na}$ (**D**) and $I_{Ca}$ (**E**), as well

*Figure 2 continued on next page*

*Figure 2 continued*

as the $I_{Ca}/I_{Na}$ ratio (F) in the same WT and MICU1-KO mitoplasts. Currents were measured at −80 mV. Data shown as mean ± SEM; unpaired t-test, two-tailed, n = 18–27. (G and H) ΔΨ depolarization induced by application of 5 mM EDTA in suspension of isolated mitochondria from mouse liver (G) and MEF (H). The degree of depolarization is expressed as percentage of the full depolarization induced by 1 μM FCCP. Both WT and MCU-KO mitochondria were assessed. Data shown as mean ± SEM; one-way ANOVA with Tukey test, n = 4–9. (I) ΔΨ depolarization induced by application of 5 mM EDTA in suspension of isolated mitochondria from MEF with WT and MICU1-deficient $MCU_{cx}$. WT data in panel I is the same as in panel H. The degree of depolarization is expressed as percentage of the full depolarization induced by 1 μM FCCP. Data shown as mean ± SEM; one-way ANOVA with Tukey test, n = 4–5. All superimposed current traces in a single panel are from the same mitoplast.

The online version of this article includes the following source data and figure supplement(s) for figure 2:

**Source data 1.** Dataset values for *Figure 2*.
**Figure supplement 1.** $I_{Ca}$ in MICU1–3 knockouts, and expression levels of various MCU subunits in MICU1-KO.
**Figure supplement 1—source data 1.** Raw western blot image for panel D.
**Figure supplement 1—source data 2.** Raw western blot image for panel E.
**Figure supplement 1—source data 3.** Raw western blot image for panel F.
**Figure supplement 1—source data 4.** Raw western blot image for panel G.
**Figure supplement 1—source data 5.** Dataset values for *Figure 2—figure supplement 1*.
**Figure supplement 2.** The sensitivity of $I_{Na}$ to $[Ca^{2+}]_i$ remains unchanged in MICU1-KO.
**Figure supplement 2—source data 1.** Dataset values for *Figure 2—figure supplement 2*.
**Figure supplement 3.** DRP1 does not affect the currents mediated by the $MCU_{cx}$ or their phenotype in MICU1-KO.
**Figure supplement 3—source data 1.** Dataset values for *Figure 2—figure supplement 3*.

in MICU1-KO cells did not rescue the $I_{Ca}$ reduction (*Figure 2—figure supplement 1G–I*). Therefore, the $I_{Ca}$ reduction in MICU1-KO was not caused by reduction in MCU or EMRE.

To better understand the role played by the MICU1 subunit in the function of the $MCU_{cx}$, we used a novel test to characterize the $MCU_{cx}$ channel properties independent of $Ca^{2+}$ conductance. We used $Na^+$ current via $MCU_{cx}$ ($I_{Na}$) in the absence of both $Ca^{2+}$ and $Mg^{2+}$ (using EDTA) (*Fieni et al., 2012*; *Garg and Kirichok, 2019*; *Kirichok et al., 2004*), to calibrate $I_{Ca}$ and characterize $MCU_{cx}$ as shown in *Figure 2C–D*. Here, it is shown that the $Na^+$ current through $MCU_{cx}$ is indistinguishable in magnitude when it is measured in WT and MICU1-KO mitoplasts (*Figure 2D*). This provides evidence that the $MCU_{cx}$ conductance pathway is the same in WT and MICU1-KO. Nevertheless, when $Ca^{2+}$ permeates $MCU_{cx}$, the $I_{Ca}$ in WT is roughly twice that of the current through MICU1-KO mitoplasts (*Figure 2E*). Also, the $I_{Ca}/I_{Na}$ ratio as measured in the same mitoplast decreased approximately twice in MICU1-KO in comparison to WT (*Figure 2F*). Importantly, the reduction in $I_{Ca}/I_{Na}$ ratio in MICU1-KO could not be explained by altered relative affinities for $Ca^{2+}$ and $Na^+$ binding in the selectivity filter, because $I_{Na}$ was inhibited to the same extent by 2 nM $[Ca^{2+}]_i$ in both WT and MICU1-KO mitoplasts (*Figure 2—figure supplement 2A and B*). From this we conclude that the MICU1 subunit enhances the MCU current at high $[Ca^{2+}]_i$ and does not occlude the $MCU_{cx}$ channel when $[Ca^{2+}]_i$ is low.

We also reproduced these results in MEFs with intact DRP1 (*Dnm1l$^{+/+}$*). In these cells, the amplitudes of $I_{Ca}$ and $I_{Na}$ were the same as in *Dnm1l$^{-/-}$* MEFs (*Figure 2—figure supplement 3A and B*). Similar to MICU1 knockout in *Dnm1l$^{-/-}$* MEFs, MICU1 knockout in *Dnm1l$^{+/+}$* MEFs did not affect $I_{Na}$ while markedly reduced $I_{Ca}$ (*Figure 2—figure supplement 3C–E*). Additionally, MICU1-KO reduced the $I_{Ca}/I_{Na}$ ratio, as measured in the same mitoplast, to the similar extent in *Dnm1l$^{+/+}$* MEFs (*Figure 2—figure supplement 3F*). Thus, as expected, Drp1 presence or absence does not affect currents mediated by the MCU complex or the MICU1-KO phenotypes.

The lack of the $MCU_{cx}$ occlusion by MICU1 at low $[Ca^{2+}]_i$ was further tested in intact isolated mitochondria as shown in *Figure 2G–I*. In these experiments, we found that depletion of $Ca^{2+}$ and $Mg^{2+}$ using EDTA enables a $Na^+$ influx via $MCU_{cx}$ that depolarizes $\Delta\Psi_m$ (*Figure 2G and H*). As shown in *Figure 2I*, this influx depolarizes $\Delta\Psi_m$ to the same extent whether MICU1 was expressed or not, again showing the lack of $MCU_{cx}$ occlusion by MICUs.

Since in MICU1-KO, all MICUs are removed from the $MCU_{cx}$ complex, we conclude that MICUs do not plug the $MCU_{cx}$ channel when $[Ca^{2+}]_i$ is low. Instead, the function of MICUs is to potentiate $MCU_{cx}$ activity at elevated $[Ca^{2+}]_i$.

## Role of the Ca$^{2+}$-binding EF hands of MICUs

The Ca$^{2+}$-dependent potentiation of MCU$_{cx}$ imparted by the MICU subunits is likely to be mediated by Ca$^{2+}$ binding to their EF hands. To test this hypothesis, we recombinantly expressed MICU1–3 or MICU1–3 with mutated EF hands (mut-EF-MICU, to disable Ca$^{2+}$ binding *Kamer et al., 2017*) in their respective knockout cell lines [*Figure 3A*] and examined the changes in $I_{Ca}$. Expression levels of both the recombinant WT and mut-EF-MICU proteins were significantly higher as compared to endogenous MICUs expression in each case (*Figure 3A*).

In MICU1-KO, expression of MICU1 was able to restore $I_{Ca}$ to the WT level, but mut-EF-MICU1 expression failed to do so (*Figure 3B*). This confirms our hypothesis that Ca$^{2+}$ binding to the EF hands of MICU1 is indispensable for the $I_{Ca}$ potentiation.

In MICU2-KO, $I_{Ca}$ was not significantly affected (*Figure 3C*, and *Figure 2A and B*), because the loss of MICU2 appeared to be compensated with increased MICU1 expression and formation of MICU1 homodimers (*Patron et al., 2014*; *Figure 3—figure supplement 1A*). Therefore, overexpression of recombinant MICU2 in the knockout background only reverted the MICU1 homodimer back to heterodimer without any change in the $I_{Ca}$ amplitude (*Figure 3C*). In contrast, mut-EF-MICU2 overexpression displaced MICU1 from the homodimers in favor of MICU1/mut-EF-MICU2 heterodimer, leading to a decrease in the total number of functional EF hands in the heterodimer. This results in a significant decrease in MICU-dependent $I_{Ca}$ potentiation (*Figure 3C*). These functional data, combined with biochemical/structural evidence for preferential formation of MICU1/MICU2 heterodimers (*Fan et al., 2020*; *Patron et al., 2014*; *Petrungaro et al., 2015*; *Wang et al., 2019*; *Wang et al., 2020a*; *Wang et al., 2020b*; *Xing et al., 2019*; *Zhuo et al., 2021*), suggest that MICU2, along with MICU1, is responsible for allosteric potentiation of MCU upon binding of cytosolic Ca$^{2+}$ to their EF hands.

The composition of MICU dimers can also be affected by MICU3 that, similar to MICU2, was proposed to interact and form heterodimers with MICU1 (*Patron et al., 2019*; *Plovanich et al., 2013*). MICU3 is a minor protein as compared to MICU1 and 2 in the majority of tissues and cell lines (*Patron et al., 2019*), which also appears to be the case in our system (*Figure 3A*). Accordingly, $I_{Ca}$ was not affected in MICU3-KO mitoplasts, and overexpression of recombinant MICU3 or mut-EF-MICU3 in MICU3-KO also had no effect on $I_{Ca}$ (*Figure 3D*). MICU3 is profoundly expressed in neurons where it was shown to increase the efficiency of mitochondrial Ca$^{2+}$ uptake in axons (*Ashrafi et al., 2020*).

Ca$^{2+}$ binding to the EF hands of MICU subunits and a subsequent conformational change that potentiates the MCU$_{cx}$ activity require a finite time and may delay $I_{Ca}$ activation/deactivation in response to rapid changes in [Ca$^{2+}$]$_i$. Therefore, we examined $I_{Ca}$ activation and deactivation kinetics in response to rapid changes in [Ca$^{2+}$]$_i$ and tested whether they depend on MICUs. $I_{Ca}$ activation upon rapid elevation of [Ca$^{2+}$]$_i$ from virtually Ca$^{2+}$-free to 1 mM was immediate, with kinetics comparable to the rate of solution exchange (τ ~0.4 ms) achieved by our fast application system (*Figure 3E*, and *Figure 3—figure supplement 2*). Importantly, the kinetics of the $I_{Ca}$ rapid response was not altered in MICU1-KO (*Figure 3E and F*). The deactivation kinetics was similarly fast and not dependent on MICU1 (*Figure 3E and F*). The result of these experiments correspond to the previous observation that EF hands of calmodulin bind Ca$^{2+}$ with a μs time constant (*Faas et al., 2011*). The conclusion from these experiments is that the kinetics of Ca$^{2+}$ binding to the MICU's EF hands, and the resultant conformational change in the MCU$_{cx}$, are fast enough that MICUs and mitochondria will rapidly track changes in [Ca$^{2+}$]$_i$.

The MCU$_{cx}$ is an inward rectifying Ca$^{2+}$ channel (*Kirichok et al., 2004*). However, it remains unclear if the MICUs contribute to this feature. To examine this possibility, we measured $I_{Ca}$ in the presence of 2 mM [Ca$^{2+}$]$_m$ (pipette solution). Under these conditions, no outward $I_{Ca}$ was observed either before or after [Ca$^{2+}$]$_i$ elevation in either WT or MICU1-KO. However, as expected, 1 mM [Ca$^{2+}$]$_i$ induced a robust inward $I_{Ca}$ (*Figure 3G*). Thus, the MICUs do not appear to be responsible for the inward rectification of MCU$_{cx}$, and the inward rectification is an inherent property of the pore proteins.

Recently published work suggested that MCU$_{cx}$ might be inhibited by matrix [Ca$^{2+}$] (*Vais et al., 2016*; *Vais et al., 2020*), specifically at [Ca$^{2+}$]$_m$ ~400 nM. However, as shown in *Figure 3—figure supplement 1B–C*, $I_{Ca}$ amplitude is unchanged when [Ca$^{2+}$]$_m$ was set at either Ca$^{2+}$-free, or 400 nM,

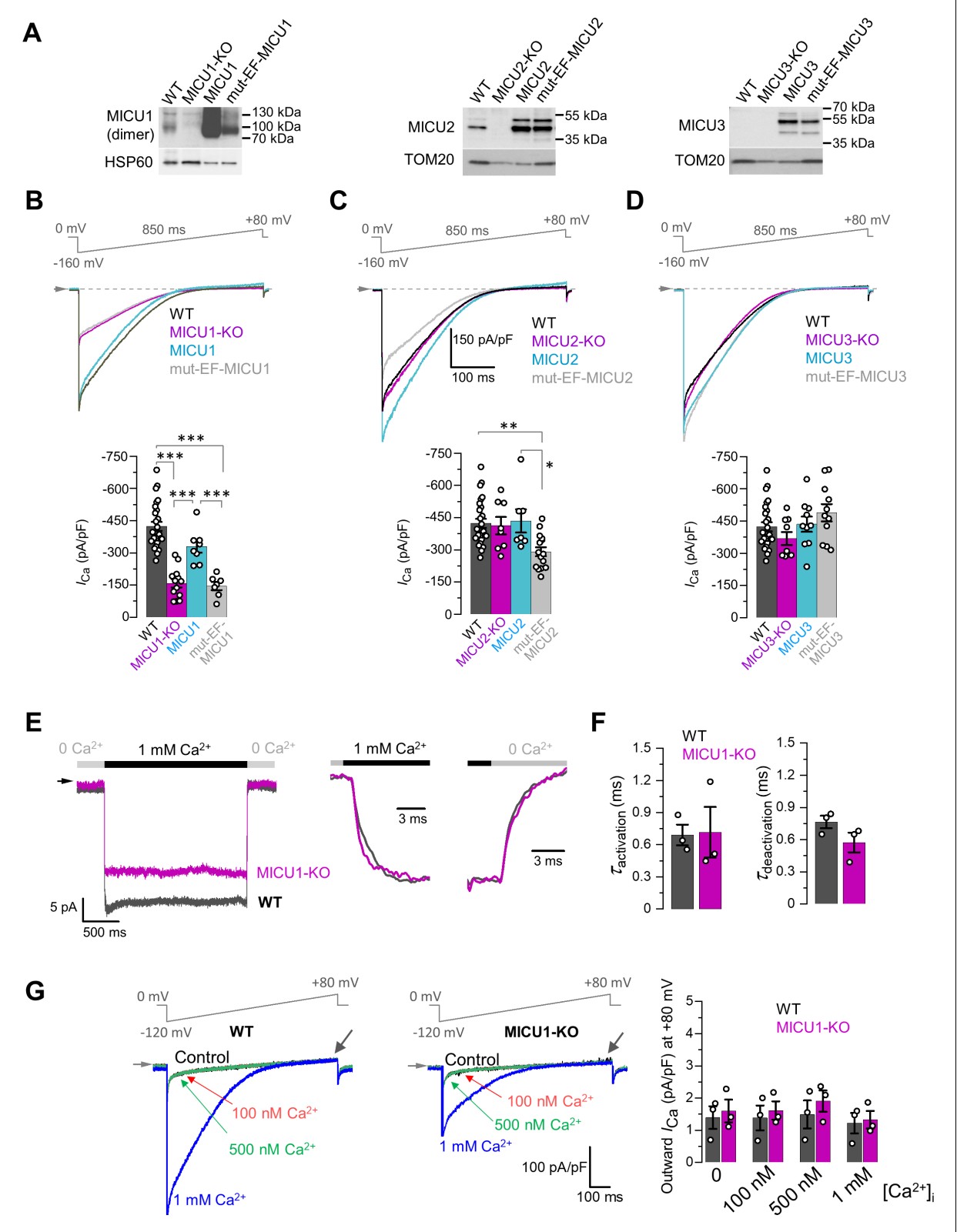

**Figure 3.** Effects of MICU subunits and their EF hands on the amplitude, kinetics and rectification of $I_{Ca}$. (**A**) Western blots showing overexpression of MICU subunits or MICU subunits with non-functional EF hands (mut-EF-MICU) in their respective knockout background (*left*, MICU1-KO; *middle*, MICU2-KO and; *right*, MICU3-KO). For MICU1 (*left panel*), samples were prepared without reducing agent, β-mercaptoethanol. The MICU1 band is near the expected molecular weight for the homo- or heterodimer (with MICU2 or 3). (**B–D**) *Upper panels*: $I_{Ca}$ in MICU1-KO (**B**), MICU2-KO (**C**), and MICU3-

*Figure 3 continued on next page*

*Figure 3 continued*

KO (D) before and after overexpression of a corresponding MICU subunit or its EF hand mutant, as compared to WT. Representative $I_{Ca}$ traces recorded from the mitoplasts of different backgrounds in 1 mM $[Ca^{2+}]_i$ are shown together in a single panel. *Lower panels:* quantification of $I_{Ca}$ amplitudes from the upper panel at −160 mV. The same WT and knockout data were used as in *Figure 2B*. All superimposed current traces in a panel are compiled from multiple mitoplasts. Data shown as mean ± SEM; one-way ANOVA with post-hoc Tukey test. n = 7–26. (E) *Left panel:* $I_{Ca}$ measured at a holding voltage of −100 mV while $[Ca^{2+}]_i$ was rapidly (τ ~0.4 ms, see Materials and methods) switched from virtual zero to 1 mM and then back to virtual zero in WT (*gray*) and MICU1-KO (*purple*) mitoplasts. *Right panel,* $I_{Ca}$ kinetics within ~10 ms after the fast $[Ca^{2+}]_i$ elevation and subsequent decrease in WT (*gray*) and MICU1-KO (*purple*) mitoplasts from the left panel. $I_{Ca}$ traces were normalized to the maximal amplitude to facilitate comparison of kinetics in WT and MICU1-KO. (F) *Left:* $I_{Ca}$ activation time constant ($\tau_{activation}$) in WT and MICU1-KO; *Right:* $I_{Ca}$ deactivation time constant ($\tau_{deactivation}$) in WT and MICU1-KO. Data shown as mean ± SEM, n = 3. (G) $I_{Ca}$ at $[Ca^{2+}]_m$ = 2 mM and indicated $[Ca^{2+}]_i$ in WT and MICU1-KO. Black arrows point out where the amplitude of outward $I_{Ca}$ was measured. Bar-graph shows the amplitude of outward $I_{Ca}$ measured at +80 mV. All superimposed current traces in a single panel are from the same mitoplast. Data shown as mean ± SEM, n = 3, each $[Ca^{2+}]_i$.

The online version of this article includes the following source data and figure supplement(s) for figure 3:

**Source data 1.** Raw western blot image for panel A.
**Source data 2.** Dataset values for *Figure 3*.
**Figure supplement 1.** Matrix $Ca^{2+}$ does not regulate $I_{Ca}$.
**Figure supplement 1—source data 1.** Raw western blot image for panel A (*Upper*).
**Figure supplement 1—source data 2.** Raw western blot image for panel A (*Lower*).
**Figure supplement 1—source data 3.** Dataset values for *Figure 3—figure supplement 1*.
**Figure supplement 2.** Fast solution stepping with the solution exchange system.

or 400 µM. Thus, the MCU$_{cx}$ is not regulated by matrix $Ca^{2+}$, and MICUs only impart the regulation of the MCU$_{cx}$ by cytosolic $Ca^{2+}$.

Taken together, these data indicate that binding of cytosolic $Ca^{2+}$ to EF hands of MICU subunits allosterically potentiates MCU$_{cx}$ currents.

## MICUs regulate the open state probability of MCU$_{cx}$ channel

To investigate the mechanism by which $Ca^{2+}$-bound MICU subunits potentiate $I_{Ca}$, we examined the activity of single MCU$_{cx}$ channels in inside-out (matrix-side out) IMM patches (*Figure 4*). Because the unitary MCU$_{cx}$ current ($i_{Ca}$ via a single MCU$_{cx}$ channel) is very small if measured in physiological $[Ca^{2+}]_i$, it must be recorded at high $[Ca^{2+}]_i$ = 105 mM to enable proper resolution (*Kirichok et al., 2004*). As desired for this experiment, using this $[Ca^{2+}]_i$, the EF hand domains of the MICU subunits are fully saturated with $Ca^{2+}$.

MCU$_{cx}$ exhibits multiple levels of single channel conductance (*Figure 4A–C*; *Kirichok et al., 2004*). The sub-conductances can be observed at all tested voltages (−40,−80, and −120 mV), but their resolution improves as the transmembrane voltage, and the amplitude of single-channel currents increase. It is clear that the open probability ($P_o$) of the MCU$_{cx}$ is increased by the hyperpolarization of the IMM as was also shown previously (*Kirichok et al., 2004*). At −120 mV there are full-sized stochastic openings of MCU$_{cx}$ as well as sub-conductance openings at ~80% and ~60% of the amplitude of the fully open $i_{Ca}$ (*Figure 4A–C*). Because similar amplitude levels were observed in all the patches, we conclude that these events represent genuine sub-conductances in the MCU$_{cx}$ channel.

There was no difference in the single channel amplitude between control and MICU1-KO mitoplasts (*Figure 4C*). However, we found that the single-channel open probability ($P_o$) was significantly decreased ~2–3 fold in MICU1-KO versus WT mitoplasts, depending on the transmembrane voltage (*Figure 4D*). As a result, the time-averaged current contributed by a single MCU$_{cx}$ channel differs significantly between control and MICU1-KO mitoplasts (*Figure 4E*), thus mirroring and explaining the effect of MICU1 knockout on the amplitude of the whole mitoplast $I_{Ca}$ (*Figure 2B*).

We next recorded MCU$_{cx}$ single channel activity using Na$^+$ as the permeating ion ($i_{Na}$), in nominally $Ca^{2+}$-free conditions (MICUs in $Ca^{2+}$-free state) (*Figure 5A and B*). Similar to our findings with $Ca^{2+}$ as the permeant ion, there were multiple conductance states (i.e. sub-conductances) when Na$^+$ was the permeant ion. These sub-conductance states were the same in WT and MICU1-KO (*Figure 5C*). However, in contrast to $i_{Ca}$, there were no significant differences in the open probability of $i_{Na}$ between WT and MICU1-KO when Na$^+$ was the permeating ion (*Figure 5D*). Accordingly, there was no difference in the time-averaged currents contributed by a single MCU$_{cx}$ channel in

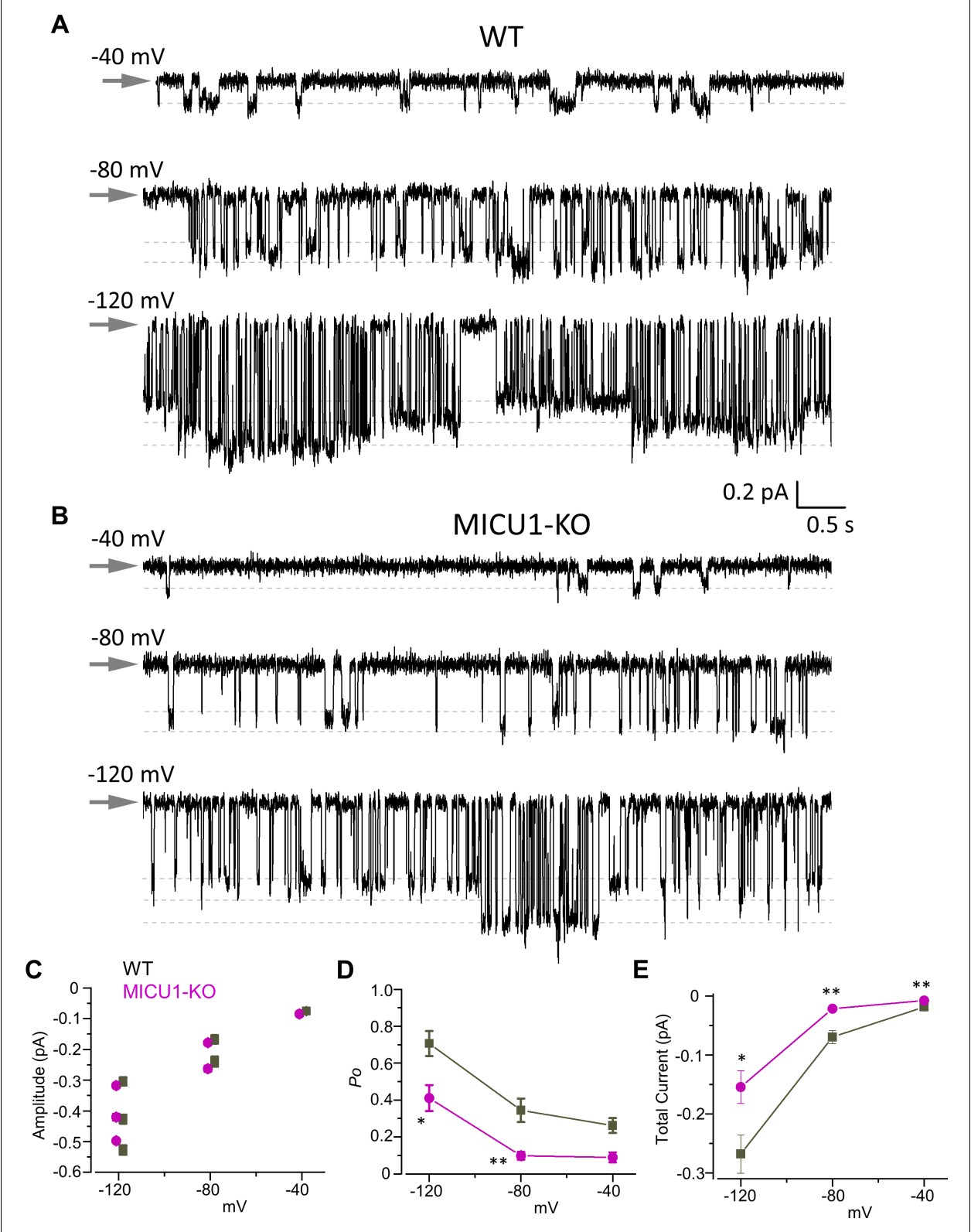

**Figure 4.** Open probability of the MCU channel in the presence of cytosolic Ca$^{2+}$ is decreased in MICU1-KO. (**A and B**) MCU single-channel Ca$^{2+}$ currents ($i_{Ca}$) from inside-out IMM patches in WT (**A**) and MICU1-KO (**B**), recorded at indicated potentials in symmetrical 105 mM Ca$^{2+}$, and low-pass filtered at 0.3 kHz for display purposes. Arrows indicate closed-state level, and downward deflections are the open-state events. Multiple

*Figure 4 continued on next page*

*Figure 4 continued*

subconductance levels are clearly visible at −80 and −120 mV. (C–E) Single-channel amplitudes (C), open probability ($P_o$) (D), and time-averaged unitary current (E) (see Methods) in WT and MICU1-KO at indicated potentials. Data shown as mean ± SEM; unpaired t-test, two-tailed, n = 5–6.

The online version of this article includes the following source data for figure 4:

**Source data 1.** Dataset values for *Figure 4*.

control and MICU1-KO mitoplasts (*Figure 5E*). This correlates well with the absence of differences in amplitude of the whole mitoplast MCU currents when $Na^+$ was the permeant ion for these two genotypes (*Figure 2D*).

These results demonstrate that in the $Ca^{2+}$-bound state, the MICUs potentiate MCU current by increasing the open probability of the MCU/EMRE pore. In the absence of $Ca^{2+}$, the MICUs do not appear to affect the pore activity.

## MCU$_{cx}$ Mn$^{2+}$ conductance

While manganese ($Mn^{2+}$) is essential for the proper function of several mitochondrial enzymes, its excessive accumulation inhibits oxidative phosphorylation and causes toxicity (*Gunter and Pfeiffer, 1990*). MCU$_{cx}$ appears to be the primary entry pathway for $Mn^{2+}$ entry into mitochondria (*Gunter et al., 2010*). Recently, it has been suggested that MICU1 is responsible for the relatively low permeability of MCU$_{cx}$ for $Mn^{2+}$ as compared to $Ca^{2+}$, and when MICU1 deficiency or loss-of-function occurs, it can lead to excessive mitochondrial $Mn^{2+}$ accumulation and cellular toxicity (*Kamer et al., 2018*; *Wettmarshausen et al., 2018*).

We recorded the current carried by $Mn^{2+}$ through MCU$_{cx}$ ($I_{Mn}$) in the presence of 5 mM $[Mn^{2+}]_i$. $I_{Mn}$ disappeared in MCU-KO and EMRE-KO, confirming that $Mn^{2+}$ current was solely mediated by MCU$_{cx}$ (*Figure 6A–C*). $I_{Mn}$ was significantly smaller (~7-fold) than $I_{Ca}$ via MCU$_{cx}$, as was also shown previously (*Kirichok et al., 2004*; *Figure 6D and E*). Interestingly, in MICU1-KO, $I_{Mn}$ and $I_{Ca}$ were reduced to a similar extent (*Figure 6F–H*). Moreover, even the ratio between $I_{Mn}$ and $I_{Ca}$ calculated from the same mitoplast ($I_{Mn}/I_{Ca}$) was not affected in MICU1-KO (*Figure 6I*). Two important conclusions follow from these observations. First, MICU1 does not differentially regulate $I_{Mn}$ and $I_{Ca}$. Second, MICU1 potentiates MCU$_{cx}$ in the presence of both $Mn^{2+}$ and $Ca^{2+}$. These results are in contrast to a popular model in which MICUs occlude MCU$_{cx}$, and that this occlusion is relieved only by $Ca^{2+}$ but not by $Mn^{2+}$ (*Kamer et al., 2018*; *Wettmarshausen et al., 2018*). However, our results are in agreement with important earlier studies which found that cytosolic $Mn^{2+}$ allosterically stimulates mitochondrial $Ca^{2+}$ uptake just like $Ca^{2+}$ (*Allshire et al., 1985*; *Hughes and Exton, 1983*; *Kröner, 1986*; *Vinogradov and Scarpa, 1973*).

We further sought to explain why $I_{Mn}$ via MCU$_{cx}$ is smaller than $I_{Ca}$. In the presence of $Mn^{2+}$, $I_{Ca}$ was decreased (*Figure 6E*). This decrease was the same in MICU1-KO demonstrating that it was a pore effect (*Figure 6F and J*). This suggests that $Mn^{2+}$ slows down $Ca^{2+}$ permeation simply because it dwells in the pore longer than $Ca^{2+}$ due to tighter binding (*Lansman et al., 1986*). The higher affinity of $Mn^{2+}$ to the pore and the longer dwell time also explains why $I_{Mn}$ is smaller than $I_{Ca}$.

Thus, the $I_{Ca}$ and $I_{Mn}$ phenotypes of MICU1-KO are the same, and MICU1 does not determine the preference of MCU$_{cx}$ for $Ca^{2+}$ over $Mn^{2+}$. Permeation of both $Ca^{2+}$ and $Mn^{2+}$ is enhanced, rather than inhibited by MICU1.

## Mg$^{2+}$ occludes the MCU$_{cx}$ pore independently of MICU1

$Mg^{2+}$, the most abundant cytosolic divalent ion, is an important negative regulator of MCU$_{cx}$-mediated mitochondrial $Ca^{2+}$ uptake (*Gunter et al., 2010*; *Hutson et al., 1976*). However, the mechanism of the inhibitory action of $Mg^{2+}$ on MCU$_{cx}$ is poorly understood. Our previous study suggested that in $Ca^{2+}$-free conditions, $Mg^{2+}$ occludes the MCU$_{cx}$ pore for $Na^+$ permeation (*Kirichok et al., 2004*). Here, we investigate how $Mg^{2+}$ affects $Ca^{2+}$ conduction through MCU$_{cx}$.

We first studied how cytosolic $Mg^{2+}$ affects activation of $I_{Ca}$ by $[Ca^{2+}]_i$ (*Figure 7A and B*). The results of these experiments clearly demonstrated that $I_{Ca}$ is inhibited in the presence of $Mg^{2+}$, and that this inhibition was primarily prominent in the lower range of micromolar $[Ca^{2+}]_i$ (*Figure 7B*). Thus, we specifically tested the effect of $[Mg^{2+}]_i$ in this range of $[Ca^{2+}]_i$ by recording $I_{Ca}$ at 30 μM $[Ca^{2+}]$. In these experiments, we found that $I_{Ca}$ in the WT MCU$_{cx}$ remains about double the $I_{Ca}$ in

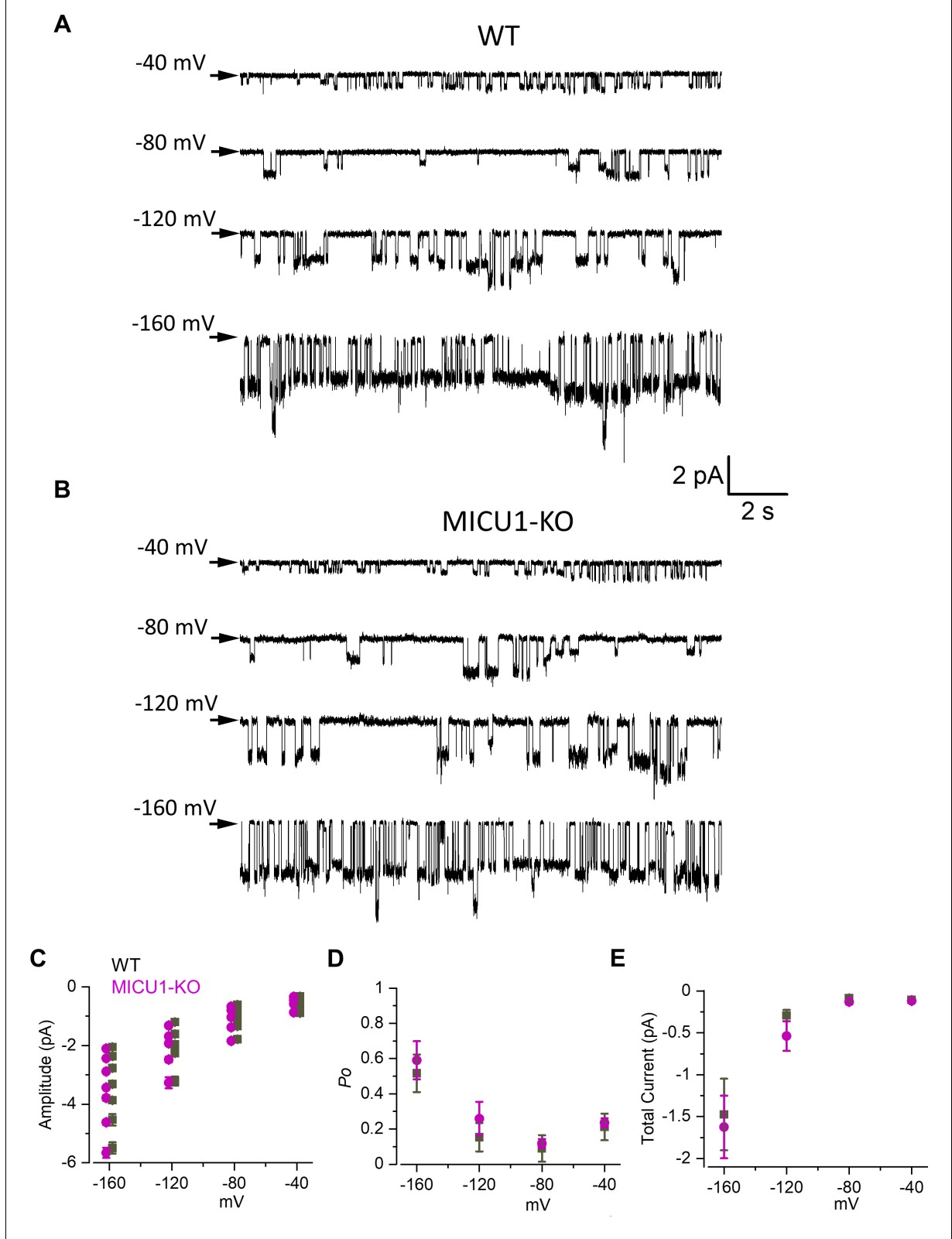

**Figure 5.** Open probability of the MCU channel in the absence of $Ca^{2+}$ remains unchanged in MICU1-KO. (**A and B**) MCU single-channel $Na^+$ currents ($i_{Na}$) from inside-out IMM patches in WT (**A**) and MICU1-KO (**B**), recorded at indicated potentials in symmetrical 150 mM $Na^+$, and low-pass filtered at 0.3 kHz for display purposes. Arrows indicate closed-state level, and downward deflections are the open-state events. Multiple subconductance levels

*Figure 5 continued on next page*

*Figure 5 continued*
are clearly visible at all potentials. (**C–E**) Single-channel amplitudes (**C**), open probability ($P_o$) (**D**), and time-averaged unitary current (**E**) (see Methods) in WT and MICU1-KO at indicated potentials. Data shown as mean ± SEM; unpaired t-test, two-tailed, n = 6–7.

The online version of this article includes the following source data for figure 5:

**Source data 1.** Dataset values for *Figure 5*.

MICU1-KO whether [$Mg^{2+}$] is 200 µM or 0 (*Figure 7C*). Thus, these data suggest that the potentiating effect that MICU1 exerts on $MCU_{cx}$ does not depend on [$Mg^{2+}$]. Furthermore, as shown in *Figure 7D*, $Mg^{2+}$ inhibits $I_{Ca}$ with similar $IC_{50}$ in WT (149 ± 20 µM) and MICU1-KO (156 ± 20 µM). Taken together, these findings indicate that $Mg^{2+}$ exerted its inhibitory effect at the $MCU_{cx}$ pore (*Figure 7C and D*) and not through the MICUs. The pronounced competitive nature of the $Mg^{2+}$ inhibition (*Figure 7B*) suggests that $Mg^{2+}$ binds within the $MCU_{cx}$ selectivity filter formed by Asp and Glu residues (*Baradaran et al., 2018*; *Fan et al., 2018*; *Nguyen et al., 2018*; *Yoo et al., 2018*). $Mg^{2+}$ is a smaller divalent cation than $Ca^{2+}$, and it is more difficult for $Mg^{2+}$ to shed its hydration shell to fit into a narrow high-affinity $Ca^{2+}$ binding site formed by Glu (site 2). However, $Mg^{2+}$ could bind to the outermost and wider Asp binding site (site 1) of the selectivity filter even with a hydration shell. Such $Mg^{2+}$ binding would not allow $Mg^{2+}$ permeation, but would occlude the pore.

In conclusion, $Mg^{2+}$ is an $MCU_{cx}$ pore blocker that at the resting [$Ca^{2+}$]$_i$, would strongly compete with $Ca^{2+}$ for binding to the selectivity filter, limiting $Ca^{2+}$ permeation. The $Mg^{2+}$ occlusion of the $MCU_{cx}$ pore can at least partially explain low mitochondrial $Ca^{2+}$ uptake at resting [$Ca^{2+}$]$_i$.

## Discussion

In summary, we demonstrate that the primary function of MICU subunits is to potentiate the activity of the $MCU_{cx}$ as cytosolic $Ca^{2+}$ is elevated and binds to MICU's EF hands. This potentiation would result in efficient stimulation of the mitochondrial ATP production in response to cytosolic $Ca^{2+}$ signaling events, when energy demand is increased as shown in neurons (*Ashrafi et al., 2020*) and heart (*Wescott et al., 2019*). Although at low [$Ca^{2+}$]$_i$, MICU1-KO mitochondria appear to have higher $Ca^{2+}$ uptake in comparison to WT, we find no evidence of a plug that blocks ion permeation via $MCU_{cx}$. On the other hand, at high [$Ca^{2+}$]$_i$, the uptake was lower in MICU1-KO, which we demonstrate by the patch clamp analysis is due to loss of MICU-mediated potentiation of $MCU_{cx}$. Mechanistically, at low [$Ca^{2+}$]$_i$ the $MCU_{cx}$ channel is open in a lower open probability mode, but as [$Ca^{2+}$]$_i$ is elevated, MICUs increase $MCU_{cx}$ open state probability, potentiating its activity (*Figure 6K*). MICUs are likely to achieve this effect by interacting with EMRE that is predicted to control the gating of the MCU pore (*Wang et al., 2019*; *Fan et al., 2020*; *Wang et al., 2020b*; *Zhuo et al., 2021*). We also show that the inward rectification property of $MCU_{cx}$ is independent of MICUs. Lastly, in contrast to the previous report (*Vais et al., 2016*), we found no evidence for the regulation of $MCU_{cx}$ activity by matrix [$Ca^{2+}$].

### Regulation of $MCU_{cx}$ function by cytosolic [$Ca^{2+}$]

Assuming that $K_d$ for $Ca^{2+}$ binding to MICU EF hands is ~600 nM (*Kamer et al., 2017*), MICUs would exert their potentiating effect over a broad range of physiological [$Ca^{2+}$]$_i$ that range from resting to low micromolar. By doing so, MICUs can help the $MCU_{cx}$ to overcome the mitochondrial $Ca^{2+}$ efflux machinery and to elevate [$Ca^{2+}$]$_m$ to achieve adequate stimulation of the mitochondrial ATP production. In previous reports, there has been significant inconsistency as to the proposed effect of MICU1 on mitochondrial $Ca^{2+}$ uptake at high [$Ca^{2+}$]$_i$ (*Supplementary file 1a and b*). For example, mitochondrial $Ca^{2+}$ uptake rate appears to be decreased in isolated liver mitochondria from mice following siRNA-mediated knockdown of MICU1 or MICU2 in an earlier study (*Plovanich et al., 2013*). Another study (*Csordás et al., 2013*) also showed decreased uptake rates when MICU1 was knocked down in HeLa cells and hepatocytes leading authors to propose that MICU1 contributes to the cooperative activation of $MCU_{cx}$, but the effect was mild and it disappeared in the absence of $Mg^{2+}$. On the other hand, many reports (*Kamer et al., 2018*; *Logan et al., 2014*; *Mallilankaraman et al., 2012a*; *Vais et al., 2016*) showed no change in MCU activity upon MICU1 loss at high [$Ca^{2+}$]$_i$ but only alteration in the threshold for mitochondrial $Ca^{2+}$ uptake. However, a recent study again showed a decrease in $Ca^{2+}$ uptake in isolated liver mitochondria in MICU1-KO

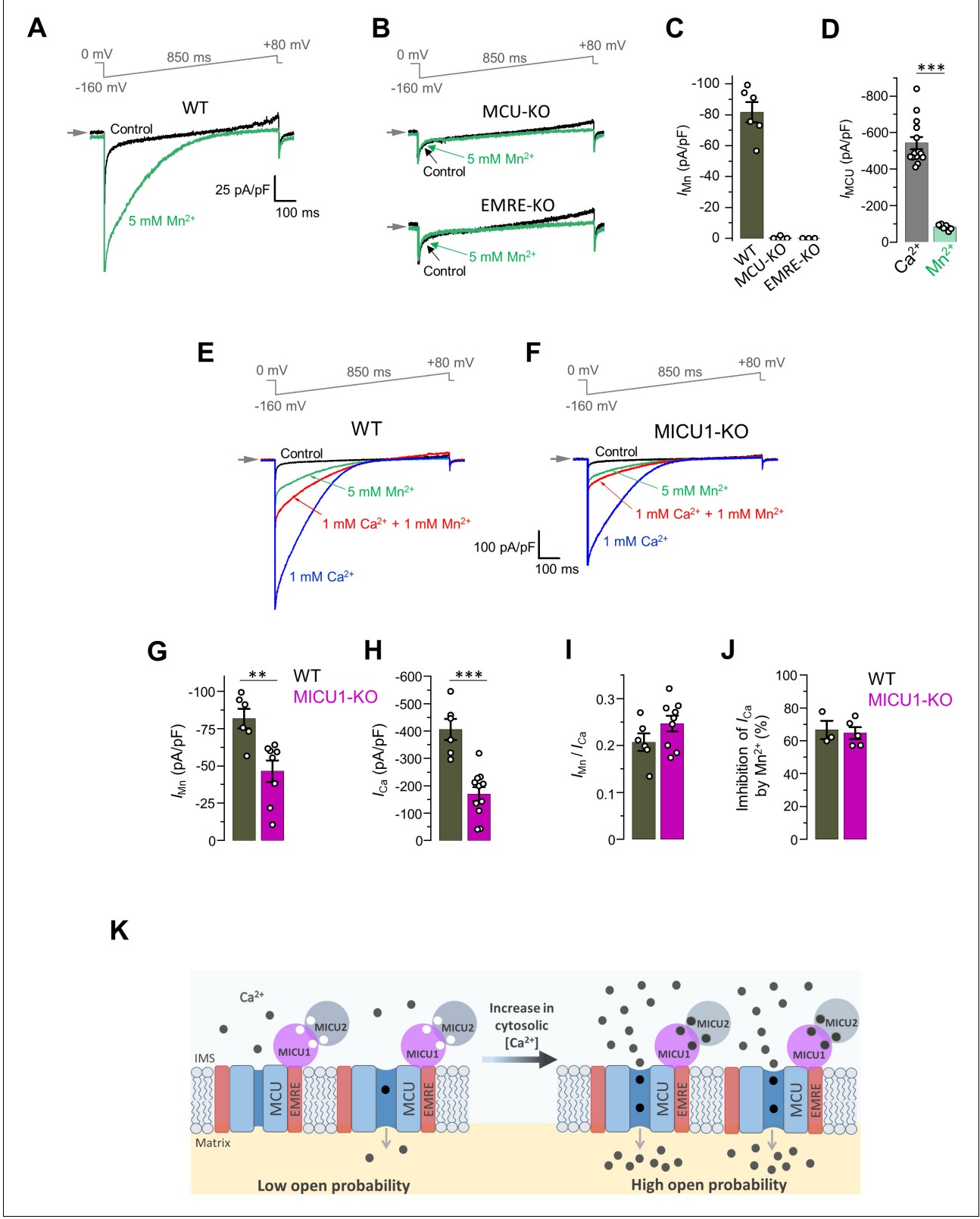

**Figure 6.** $I_{Mn}$ is reduced in MICU1-KO to the similar extent as $I_{Ca}$. (**A and B**) Representative inward $I_{Mn}$ in WT (**A**), MCU-KO (*upper panel*) and EMRE-KO (*lower panel*, **B**) mitoplasts at 5 mM $[Mn^{2+}]_i$. All superimposed current traces in a single panel are from the same mitoplast. (**C**) $I_{Mn}$ measured at −160 mV from WT, MCU-KO and EMRE-KO mitoplasts. Data shown as mean ± SEM. (n = 3–6). (**D**) MCU current amplitude ($I_{MCU}$) in the presence of 5 mM $[Ca^{2+}]_i$ and 5 mM $[Mn^{2+}]_i$ in WT mitoplasts. Currents were measured at −160 mV. Data shown as mean ± SEM; unpaired t-test, two-tailed. n = 6–14. (**E**

*Figure 6 continued on next page*

*Figure 6 continued*
and F) Representative $I_{Ca}$ (*blue*, $[Ca^{2+}]_i$ = 1 mM), $I_{Mn}$ (*green*, $[Mn^{2+}]_i$ = 5 mM) and inhibition of $I_{Ca}$ by $Mn^{2+}$ (*red*, $[Ca^{2+}]_i$ = 1 mM and $[Mn^{2+}]_i$ = 1 mM) as recorded from the same mitoplast in WT (**E**) and MICU1-KO (**F**). All superimposed current traces in a single panel are from the same mitoplast. (**G–J**) $I_{Mn}$ (**G**), $I_{Ca}$ (**H**), $I_{Mn}/I_{Ca}$ ratio (**I**, measured in the same mitoplast), and inhibition of $I_{Ca}$ by 1 mM $[Mn^{2+}]_i$ (**J**) in WT and MICU1-KO. Data shown as mean ± SEM; unpaired t-test, two-tailed, n = 3–11. (**K**) Proposed model of the MCU complex gating and the role of MICU subunits in $Ca^{2+}$-dependent potentiation of the MCU current. The MCU complex is a constitutively active channel and the level of its activity is determined by the probability of open state ($P_o$). At resting $[Ca^{2+}]_i$, $P_o$ is low. As $[Ca^{2+}]_i$ is increased and $Ca^{2+}$ binds to the EF hands of MICU subunits, MICUs increase $P_o$, resulting in the increase in the MCU activity.

The online version of this article includes the following source data for figure 6:

**Source data 1.** Dataset values for *Figure 6*.

mouse relative to WT (*Liu et al., 2016*). This correlates well with our data showing direct potentiation of the $MCU_{cx}$ activity by MICU1. Thus, the potentiating effect of MICUs on the $MCU_{cx}$ was discernible in the previous research but was largely rejected due to the predominant view that the primary function of MICUs is to occlude the $MCU_{cx}$ pore.

Our data is incompatible with the model in which MICUs occlude the MCU pore at low $[Ca^{2+}]_i$ and impart a $[Ca^{2+}]_i$ activation threshold on the $MCU_{cx}$. This model explains the increase of

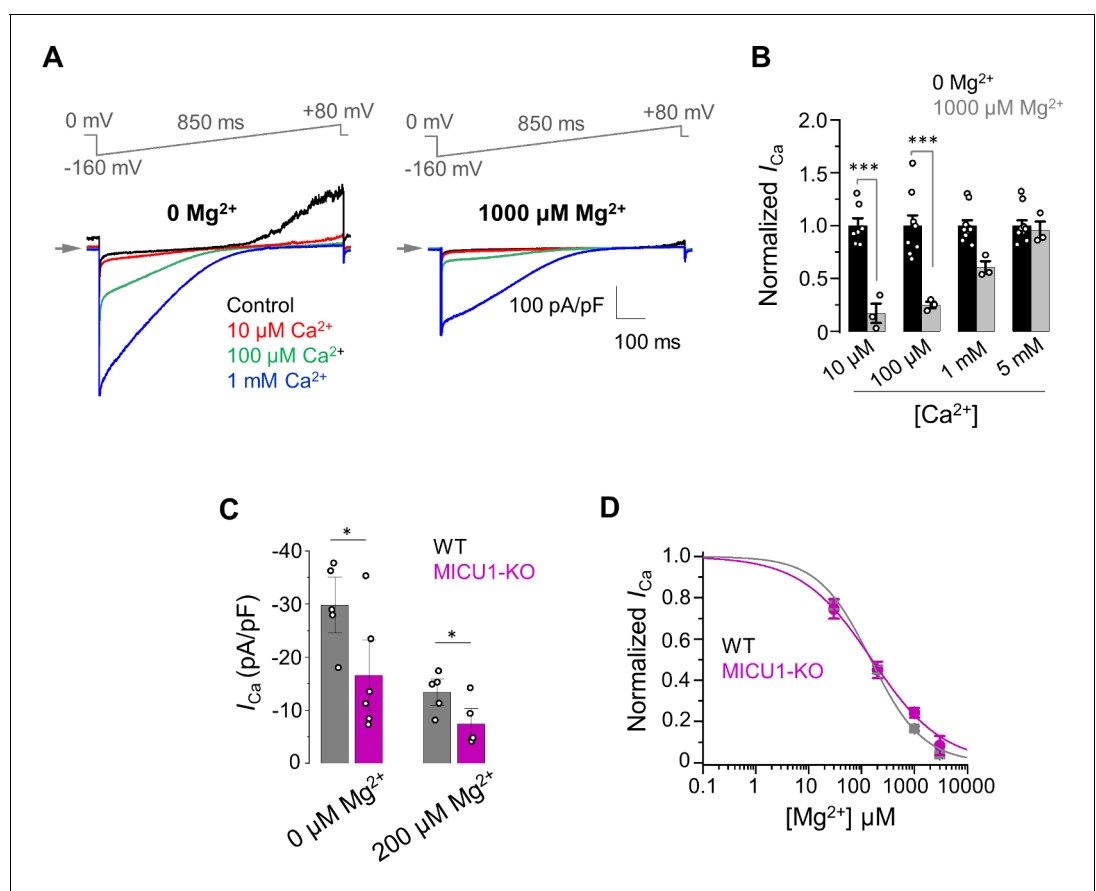

**Figure 7.** The inhibitory effect of $Mg^{2+}$ on $MCU_{cx}$ does not depend on MICU1. (**A**) Inward $I_{Ca}$ elicited at indicated $[Ca^{2+}]_i$ in the presence of 0 (*left*) and 1000 μM (*right*) $[Mg^{2+}]_i$ in WT mitoplasts. (**B**) $I_{Ca}$ elicited at indicated $[Ca^{2+}]_i$ in the presence of 0 and 1000 μM $[Mg^{2+}]_i$. $I_{Ca}$ amplitudes were measured at −160 mV in WT mitoplasts and normalized per $I_{Ca}$ at $[Mg^{2+}]_i$ = 0. Mean ± SEM; unpaired t-test, two-tailed. (**C**) Quantification of $I_{Ca}$ elicited at 30 μm $[Ca^{2+}]_i$ in the presence of 0 and 200 μM $[Mg^{2+}]_i$ in WT and MICU1-KO. Currents were measured at −160 mV. Data shown as mean ± SEM; unpaired t-test, two-tailed, n = 5–6. (**D**) Dose-response curve of $I_{Ca}$ inhibition by $[Mg^{2+}]_i$ in WT ($IC_{50}$ = 149 ± 20 μM, $n_H$ = 0.9 ± 0.1; n = 5) and MICU1-KO ($IC_{50}$ = 156 ± 21 μM, $n_H$ = 0.7 ± 0.1; n = 4). Mean ± SEM; unpaired t-test, two-tailed.

The online version of this article includes the following source data for figure 7:

**Source data 1.** Dataset values for *Figure 7*.

mitochondrial $Ca^{2+}$ accumulation at resting $[Ca^{2+}]_i$ in MICU1-KO to 'unblocking' of $MCU_{cx}$. However, in this concentration range, the $MCU_{cx}$ activity is so slow that the mitochondrial $Ca^{2+}$ accumulation is definitely influenced by many other factors. Among other possibilities, one can speculate that mitochondrial $Ca^{2+}$ efflux, matrix $Ca^{2+}$ buffering, matrix pH and permeability of the outer mitochondrial membrane might be altered in MICU1-KO to facilitate mitochondrial $Ca^{2+}$ accumulation at low $[Ca^{2+}]_i$.

Because these multiple factors can be affected by MICU1-KO differently in different experimental system and conditions, the change in the threshold can vary from one system to another. In *Trypanosoma cruzi*, where the composition of the MCU complex (including EMRE and MICU1) is similar to mammals, MICU1 knockout causes an increase in the $Ca^{2+}$ uptake 'threshold' and a marked decrease in $Ca^{2+}$ uptake capacity at all $[Ca^{2+}]_i$ (*Bertolini et al., 2019*). A recent report also found no apparent $[Ca^{2+}]_i$ threshold for MCU in isolated mitochondria from heart and skeletal muscle (*Wescott et al., 2019*). Similarly sympathetic neurons lack a discernible threshold and mitochondria were shown to accumulate small amount of $Ca^{2+}$ at $[Ca^{2+}]_i$ levels as low as 200 nM (*Colegrove et al., 2000a*; *Colegrove et al., 2000b*). Finally, interpretation of the MICU1-KO phenotypes can be further complicated by possible functional roles of MICU1 outside of the MCU complex (*Gottschalk et al., 2019*; *Tomar et al., 2019*; *Tufi et al., 2019*). In particular, MICU1 was proposed to interact with the MICOS complex, affecting the cristae structure (*Tomar et al., 2019*).

## Examination of the conduction pathway of the $MCU_{CX}$

To circumvent the problem of isolating the $MCU_{cx}$ activity from the other mitochondrial $Ca^{2+}$ homeostatic mechanisms, we leveraged the capacity of $MCU_{cx}$ to conduct $Na^+$. The $Na^+$ permeation via $MCU_{cx}$ is observed not only in isolated mitoplasts but also in intact mitochondria. $Na^+$ permeates via $MCU_{cx}$ because $Ca^{2+}$ and $Na^+$ ions have the same ionic diameter, and $Na^+$ can pass through $Ca^{2+}$ channels when $Ca^{2+}$ is removed from the selectivity filter (*Hess and Tsien, 1984*; *Hess et al., 1986*; *Tang et al., 2014*). $Ca^{2+}$ binds to the $MCU_{cx}$ selectivity filter with an exceptionally high affinity ($K_d \leq 2$ nM) (*Kirichok et al., 2004*), and re-addition of cytosolic $Ca^{2+}$ leads to $I_{Na}$ inhibition upon $Ca^{2+}$ binding to this site. Importantly, MICUs are not involved in this inhibition, as it is not affected by MICU1-KO (*Figure 2—figure supplement 2*). In intact mitochondria, $Na^+$ uptake via WT or MICU1-deficient $MCU_{cx}$ causes the same $\Delta\Psi$ depolarization, demonstrating that at low $[Ca^{2+}]_i$ $MCU_{cx}$ is not occluded regardless of its association with MICUs. Moreover, using mitochondrial patch-clamp under $Ca^{2+}$-free conditions, we recorded a robust $I_{Na}$ via MCU, and the amplitude of this $I_{Na}$ was not affected in MICU1-KO. This demonstrates that MCU pore is not occluded by MICUs at low $[Ca^{2+}]_i$.

The complexity of the $MCU_{cx}$ presents formidable difficulties even for the direct electrophysiological analysis. The electrophysiological phenotypes observed may be associated with altered expression levels of other subunits rather than the loss of MICU1. We not only analyzed the expression levels of all MCU complex subunits (*Figure 2—figure supplement 1D–F*) but also measured $I_{Ca}/I_{Na}$ ratio in the same mitoplast (*Figure 2C–F*) to ensure that we correctly capture the role of MICU1 in the MCU complex.

## Cryo-EM studies and the dynamic function of MICUs

Recently, cryo-EM structures of the $MCU_{cx}$ holocomplex reveal that it is composed of two conjoined MCU/EMRE pores with a MICU1/MICU2 heterodimer attached to each of them (*Fan et al., 2020*; *Wang et al., 2020b*; *Zhuo et al., 2021*). Based on molecular weight of the $MCU_{cx}$ as observed on the Blue native PAGE (*Sancak et al., 2013*), this structure likely represents the complete $MCU_{cx}$ of intact mitochondria. This is the primary structural form of $MCU_{cx}$ in the mitochondrial inner membrane.

However, in addition to this complete $MCU_{cx}$, the cryo-EM analysis also reported structures of a monomeric MCU/EMRE pore with a single MICU1/MICU2 heterodimer. Although this does not appear to be the predominant state of the native $MCU_{cx}$, it was primarily used for the analysis of the interaction between the pore and the MICU1/MICU2 dimer. Based on this analysis, a conclusion was made that in the absence of $Ca^{2+}$, MICU1/MICU2 dimer occluded the $MCU_{cx}$ pore. Wang et al. (*Wang et al., 2020b*) observed occlusion of the dimeric complex, but such MICU1-occluded dimeric complex represented only ~10% of the total number of analyzed particles in the absence of $Ca^{2+}$.

The low prevalence of the dimeric complexes can be purely due to the experimental limitations, but a more thorough analysis of occlusion in this native form of MCU$_{cx}$ is warranted. At the same time, another work that analyzed the complete MCU$_{cx}$ with two pores and two MICU1/MICU2 heterodimers, did not show any occlusion (*Zhuo et al., 2021*). In this structure, the two MICU1/MICU2 heterodimers interact between each other, leaving both pores unoccluded (*Zhuo et al., 2021*). As more MCU$_{cx}$ structural data appear in the future, it is important to take into consideration the completeness of the complex and the new functional data on the MICU subunits presented here.

We are at the very beginning of the structural characterization of MCU$_{cx}$. As this crucial work continues, selection of the experimental conditions is key to understanding how the channel works under physiological conditions. As we emphasize here, the direct inhibitory effect of Mg$^{2+}$ on the MCU pore is important but has not yet been taken into consideration. In fact, Mg$^{2+}$ was omitted in all structural studies of MCU$_{cx}$ holocomplex (*Fan et al., 2020*; *Wang et al., 2020b*; *Zhuo et al., 2021*). As we discussed before, Mg$^{2+}$ is likely to bind to the same Asp ring of the selectivity filter where MICU1 was predicted to bind and, may therefore significantly affect conformational states of MCU$_{cx}$. Thus, to determine how MICU1 interacts with the MCU$_{cx}$ pore, structural studies should be performed under conditions that include physiological concentrations of Mg$^{2+}$. Furthermore, occlusion of the MCU/EMRE pore by MICUs in the absence of Ca$^{2+}$ was observed when the cryoEM particles were obtained in a solution that had ionic strength ~three times lower than physiological (*Fan et al., 2020*; *Wang et al., 2020b*). In contrast, the unoccluded state was observed at the physiological ionic strength (*Zhuo et al., 2021*). This difference is critical, because the occlusion of the MCU pore by MICU1 strongly depends on ionic interactions (*Fan et al., 2020*; *Wang et al., 2020b*). Another important consideration is the presence of cardiolipin in MCU$_{cx}$ structures, likely affected by the type of detergent used during purification. Interestingly, the occluded state was observed in the structures that contained no cardiolipin (*Fan et al., 2020*; *Wang et al., 2020b*), while the unoccluded MCU$_{cx}$ structure contained cardiolipin (*Zhuo et al., 2021*). This point is important as cardiolipin interacts with the MCU complex (*Ghosh et al., 2020*; *Zhuo et al., 2021*) and may regulate MICU function (*Kamer and Mootha, 2014*).

In conclusion, we find no evidence for the occlusion of the MCU$_{cx}$ pore by MICUs at resting [Ca$^{2+}$]$_i$. Instead, under these conditions MCU$_{cx}$-mediated uptake is limited by the low concentration of the conducting ion and by the Mg$^{2+}$ block. A slow mitochondrial Ca$^{2+}$ uptake at resting [Ca$^{2+}$]$_i$ is likely at balance with the Ca$^{2+}$ efflux mechanisms, which prevents mitochondrial Ca$^{2+}$ overload (*Nicholls, 2005*). On the other hand, the phenomenon of allosteric potentiation of MCU$_{cx}$ has been known for many decades (*Gunter et al., 2010*; *Kröner, 1986*), and this work provides its clear mechanistic explanation.

# Materials and methods

## Key resources table

| Reagent type (species) or resource | Designation | Source or reference | Identifiers | Additional information |
|---|---|---|---|---|
| Cell line (*M. musculus*) | DRP1-KO (*Dnm1l*$^{-/-}$) (Mouse embryonic fibroblast) | *Ishihara et al., 2009* | | Cell line maintained in D. Chan and K. Mihara lab; *Dnm1l*$^{-/-}$ background |
| Cell line (*M. musculus*) | WT (*Dnm1l*$^{+/+}$) (Mouse embryonic fibroblast) | *Liu et al., 2016* | | Cell line maintained in T. Finkel lab; *Dnm1l*$^{+/+}$ background |
| Cell line (*M. musculus*) | MICU1-KO (*Micu1*$^{-/-}$) (Mouse embryonic fibroblast) | *Liu et al., 2016* | | Cell line maintained in T. Finkel lab; *Dnm1l*$^{+/+}$ background |
| Cell line (*M. musculus*) | MCU-KO (*Mcu*$^{-/-}$) (Mouse embryonic fibroblast) | This paper | | Cell line maintained in Y. Kirichok and V. Garg lab. *Dnm1l*$^{-/-}$ background |
| Cell line (*M. musculus*) | EMRE-KO (*Smdt1*$^{-/-}$) (Mouse embryonic fibroblast) | This paper | | Cell line maintained in Y. Kirichok and V. Garg lab. *Dnm1l*$^{-/-}$ background |

*Continued on next page*

*Continued*

| Reagent type (species) or resource | Designation | Source or reference | Identifiers | Additional information |
|---|---|---|---|---|
| Cell line (*M. musculus*) | MICU1-KO (*Micu1$^{-/-}$*) (Mouse embryonic fibroblast) | This paper | | Cell line maintained in Y. Kirichok and V. Garg lab. *Dnm1l$^{-/-}$* background |
| Cell line (*M. musculus*) | MICU2-KO (*Micu2$^{-/-}$*) (Mouse embryonic fibroblast) | This paper | | Cell line maintained in Y. Kirichok and V. Garg lab. *Dnm1l$^{-/-}$* background |
| Cell line (*M. musculus*) | MICU3-KO (*Micu3$^{-/-}$*) (Mouse embryonic fibroblast) | This paper | | Cell line maintained in Y. Kirichok and V. Garg lab. *Dnm1l$^{-/-}$* background |
| Strain, strain background (*M. musculus*) | MCU-KO (*Mcu$^{-/-}$*) mouse | This paper | | Mouse line maintained in T. Finkel lab |
| Recombinant DNA Reagent | *Mcu* (plasmid) | This paper | | Lentiviral Construct (Y. Kirichok and V. Garg lab) |
| Recombinant DNA Reagent | *Smdt1* (alias *Emre*) (plasmid) | This paper | | Lentiviral Construct (Y. Kirichok and V. Garg lab) |
| Recombinant DNA Reagent | *Micu1* (plasmid) | This paper | | Lentiviral Construct (Y. Kirichok and V. Garg lab) |
| Recombinant DNA Reagent | *Micu2* (plasmid) | This paper | | Lentiviral Construct (Y. Kirichok and V. Garg lab) |
| Recombinant DNA Reagent | *Micu3* (plasmid) | This paper | | Lentiviral Construct (Y. Kirichok and V. Garg lab) |
| Recombinant DNA Reagent | *mut-EF-Micu1* (plasmid) | This paper | | Lentiviral Construct (Y. Kirichok and V. Garg lab) |
| Recombinant DNA Reagent | *mut-EF-Micu2* (plasmid) | This paper | | Lentiviral Construct (Y. Kirichok and V. Garg lab) |
| Recombinant DNA Reagent | *mut-EF-Micu3* (plasmid) | This paper | | Lentiviral Construct (Y. Kirichok and V. Garg lab) |
| Recombinant DNA Reagent | *Cepia2mt* (plasmid) | **Suzuki et al., 2014.** Lentiviral construct was made in this paper. | | Lentiviral construct |
| Antibody | MCU antibody (rabbit polyclonal) | Sigma | HPA016480; RRID:AB_2071893 | WB (1:2000) |
| Antibody | EMRE antibody (mouse monoclonal) | Santa Cruz | sc-86337; RRID:AB_2250685 | WB (1:200) |
| Antibody | MICU1 antibody (rabbit polyclonal) | Cell Signalling | D4P8Q (12524S); RRID:AB_2797943 | WB (1:2000) |
| Antibody | MICU2 antibody (mouse monoclonal) | Bethyl laboratories | A300-BL19212 | WB (1:500) |
| Antibody | MICU3 antibody (mouse monoclonal) | Sigma | HPA024779; RRID:AB_1848023 | WB (1:1000) |
| Antibody | VDAC antibody (rabbit monoclonal) | Santa Cruz | ab15895; RRID:AB_2214787 | WB (1:1000) |
| Antibody | TOM20 antibody (rabbit polyclonal) | Santa Cruz | sc-11415; RRID:AB_2207533 | WB (1:2000) |
| Antibody | HSP60 antibody (rabbit polyclonal) | Santa Cruz | sc-1052; RRID:AB_631683 | WB (1:3000) |
| Chemical compound, drug | ANTI-FLAG M2 Affinity Gel | Sigma-Aldrich | Cat# A2220; RRID:AB_10063035 | |

*Continued on next page*

*Continued*

| Reagent type (species) or resource | Designation | Source or reference | Identifiers | Additional information |
|---|---|---|---|---|
| Software, algorithm | PClamp 10 | Molecular Devices | | https://www.moleculardevices.com/systems/conventional-patch-clamp/pclamp-10-software |
| Software, algorithm | Origin 7.5 | OriginLab | | http://www.originlab.com/ |
| Software, algorithm | ImageJ Software | ImageJ | | https://imagej.net/ |
| Software, algorithm | [Ca$^{2+}$]$_m$ threshold detection algorithm | Custom-made | | https://github.com/ishanparanjpe/upstroke (*Paranjpe et al., 2019*) |

## Contact for reagent and resource sharing

Further information and requests for reagents may be directed to and will be fulfilled by Lead Contact Yuriy Kirichok (yuriy.kirichok@ucsf.edu).

## Experimental model

### Cell culture and recombinant gene expression

All mouse embryonic fibroblast (MEF) cells with (*Liu et al., 2016*) or without Drp1 (*Ishihara et al., 2009*), and all knockout clones were grown in low glucose (5.6 mM) Dulbecco's modified Eagle's medium (DMEM) supplemented with 10% FBS, 100 U/ml penicillin, and 100 U/ml streptomycin at 37°C, 5% $CO_2$. Cells were maintained by splitting every 48–72 hr at a ratio of 1:5 to 1:10. The MEF cell lines were authenticated by short tandem repeat profiling conducted by Labcorp. The cell lines were free of mycoplasma as determined by PCR based detection (*Dreolini et al., 2020*).

We used third-generation lentiviral (bi-cistronic) vectors containing the ORF for gene of interest with or without a selection marker (EGFP, mCherry or puromycin). The vectors were generated by VectorBuilder, Inc (Chicago, IL, USA), and their sequences were confirmed independently by the company and by us. Recombinant cDNA expressing cells were enriched using multiple rounds of FACS or antibiotic selection. In some cases, EGFP was targeted to mitochondria (using a mitochondrial targeting sequence from COX8) to identify mitoplasts expressing the recombinant protein of interest during patch clamp experiments.

## Animals

Mice were maintained on a standard rodent chow diet under 12 hr light and dark cycles. All animal experiments were performed with male mice (2–5 month old) according to procedures approved by the UCSF Institutional Animal Care and Use Committee and adhered to NIH standards. C57BL/6J were obtained from the Jackson laboratory. MCU-KO mice were obtained from Dr. Torren Finkel and have been used previously (*Pan et al., 2013*).

## Method details

Gene expression analysis (qRT-PCR) qPCR was performed by Syd Labs (Natick, MA, USA). Total RNA was isolated from cells using the RNAeasy Minikit (QIAGEN), and reverse transcribed using the First Strand cDNA Synthesis Kit (Syd Labs). qPCR reactions were performed with the following gene-specific primers (generated by Integrated DNA Technologies):

*Hprt*, Forward Primer 5'-GTCCCAGCGTCGTGATTAGC-3'
Reverse Primer 5'-GTGATGGCCTCCCATCTCCT-3'
*Mcu*, Forward Primer 5'-AAGGGCTTAGCGAGTCTTGTC-3'
Reverse Primer 5'- GGGTGCTGGTGTGTTAGTGT −3'
*Mcub*, Forward Primer 5'-CCACACCCCAGGTTTTATGTATG-3'
Reverse Primer 5'-ATGGCAGAGTGAGGGTTACCA-3'
*Smdt1*, Forward Primer 5'-ATTTTGCCCAAGCCGGTGAA-3'
Reverse Primer 5'-CCTCAAGCAGAGCAGCGAAG-3'
*Micu1*, Forward Primer 5'-CTTAACACCCTTTCTGCGTTGG-3'

Reverse Primer 5'-AGCATCAATCTTCGTTTGGTCT-3'
*Micu2*, Forward Primer 5'-CTCCGCAAACAGCGGTTCAT-3'
Reverse Primer 5'-TGCCAGCTTCTTGACCAGTG-3'
*Micu3*, Forward Primer 5'-GTAAGGTCAGAGCACGCAGAA-3'
Reverse Primer 5'-TTTCCTGTTGGACGCTGACAA −3'

cDNA (100 ng, calculated from initial RNA) samples were pre-amplified for 12 cycles using ABsolute qPCR SYBR Green Low ROX Mix (ThermoFisher). qPCR reactions were performed using an Agilent MX3000 (Fluidigm) with 40 cycles of amplification (15 s at 95℃, 5 s at 70℃, and 60 s at 60℃). Ct values were calculated by the Real-Time PCR Analysis Software (Fluidigm). Relative gene expression was determined by the ΔCt method. *Hprt* was selected as the reference gene.

## Generation of knockout cell lines by the CRISPR/Cas9 method

Knockout MEF cell lines were generated using the CRISPR/Cas9 method (*Ran et al., 2013*). All knockouts (except the MCU-KO line) were generated by Alstem LLC (Richmond, CA, USA). Either one sgRNA or a pair of two adjacent sgRNAs were used to create a point indel or a truncate indel, respectively (Figure S1).

*Mcu*, TGGCAGCGCTCGCGTCGAGA GGG
*Smdt1*, GAGTGTCCCGACATAGAGAA AGG
CTTACACTCCCACTAGGTTA AGG
*Micu1*, TCACTTTTAGATGCTGCCGG TGG
CTGCAAGTACCGGTCTCCTG TGG
*Micu2*, CGTTCGGGAGCCCTCGCGCG CGG
GGGCGCTTCCGCAAAGATGG CGG
*Micu3*, GGGCGAGCTGAGCATCGCGG CGG
CCGGGGCCGCTAGCTCCGAG GGG

MEFs were transfected with the Cas9 gRNA vector (Addgene: PX459) via electroporation (Invitrogen Neon transfection system) using the following parameters: $1 \times 10^6$ cells and 1 μg of two different gRNA-Cas9 plasmids. Puromycin was used for enrichment of transfected cells, and serial dilution was performed to select single-cell clones. A stable homozygous knockout cell line was confirmed by PCR amplification of the targeted region, cloning into a pUC19 vector, and sequencing showing that either a frameshift or large deletion had occurred in the targeted region of the gene (*Figure 1—figure supplement 1*). All knockout clones were further validated by western blotting (*Figure 1—figure supplement 1*). The primers used for amplification of genomic sites and cloning into pUC19 sequencing vector were as follows:

*Mcu*, Forward Primer TAGAAGCTTTCCACTGCTCTGATTGATCTTG
Reverse Primer ATGTGAATTCGAGCTGCTTTGGAATGAGAC
*Smdt1*, Forward Primer GTGAAGCTTGGGATCAGTAGTCCATTGGAGG
Reverse Primer AGGAGAATTCAGTGAGAGTTCCTGTGGTATG
*Micu1*, Forward Primer TTTAAGCTTGATTCCTTTGAGTTATAAGTAG
Reverse Primer CAAAGAATTCAGCAAAGAAATTCTGATGTA
*Micu2*, Forward Primer ACCAAGCTTGAACGTCGAGGAAGCAGCCAC
Reverse Primer AGGAGAATTCTCCATCCACCAGGTGGGCAG
*Micu3*, Forward Primer CGCAAGCTTCTCGCGAGATTTCGGCCCGCC
Reverse Primer AGGAGAATTCTCCATCCACCAGGTGGGCAG

## Isolation of mitochondria and mitoplasts

Mitoplasts were isolated from MEFs using methodology previously described (*Garg and Kirichok, 2019*). Briefly, MEFs were homogenized in ice-cold medium (Initial medium) containing 250 mM sucrose, 10 mM HEPES, 1 mM EGTA, and 0.1% bovine serum albumin (BSA) (pH adjusted to 7.2 with Trizma base) using a glass grinder with six slow strokes of a Teflon pestle rotating at 280 rpm. The homogenate was centrifuged at 700× g for 10 min to create a pellet of nuclei and unbroken cells. The first nuclear pellet was resuspended in the fresh Initial medium and homogenized again to increase the mitochondrial yield. Mitochondria were collected by centrifugation of the supernatant at 8500× g for 10 min.

Mitoplasts were produced from mitochondria using a French press. Mitochondria were suspended in a hypertonic solution containing 140 mM sucrose, 440 mM D-mannitol, 5 mM HEPES, and 1 mM EGTA (pH adjusted to 7.2 with Trizma base) and then subjected to a French press at 1200–2000 psi to rupture the outer membrane. Mitoplasts were pelleted at 10,500× g for 15 min and resuspended for storage in 0.5–1 ml of solution containing 750 mM KCl, 100 mM HEPES, and 1 mM EGTA (pH adjusted to 7.2 with Trizma base). Mitoplasts prepared and stored with this method contained the same amount of auxiliary MICU1 and MICU2 subunits as compared to intact mitochondria (*Figure 1—figure supplement 2F*, see TCo-immunoprecipitation section below).

Mitochondria and mitoplasts were prepared at 0–4°C and stored on ice for up to 5 hr. Immediately before the electrophysiological experiments, 15–50 µl of the mitoplast suspension was added to 500 µl solution containing 150 mM KCl, 10 mM HEPES, and 1 mM EGTA (pH adjusted to 7.0 with Trizma base) plating on 5 mm coverslips pretreated with 0.1% gelatin to reduce mitoplast adhesion.

## Patch-clamp recording

Whole mitoplast currents were measured as described previously (*Garg and Kirichok, 2019*). Giga-ohm seals with mitoplasts were formed in the bath solution containing 150 mM KCl, 10 mM HEPES and 1 mM EGTA, pH 7.2 (adjusted with KOH). Voltage steps of 350–500 mV for 2–8 ms were applied to rupture the IMM and obtain the whole-mitoplast conFiguration. Typically, pipettes had resistances of 20–40 MΩ, and the access resistance was 35–65 MΩ. The membrane capacitances of mitoplasts range from 0.2 to 0.6 pF.

All indicated voltages are on the matrix side of the IMM (pipette solution), relative to the cytosolic side (bath solution, *Figure 1—figure supplement 2D*; *Figure 1—figure supplement 2E*; *Garg and Kirichok, 2019*). Currents were normally induced by a voltage ramp from −160 mV to +80 mV (interval between pulses was 5 s) to cover all physiological voltages across the IMM, but other voltage protocols were also used as indicated in the Figures. All whole-IMM recordings were performed under continuous perfusion of the bath solution. Currents were normalized per membrane capacitance to obtain current densities (pA/pF). Currents flowing into mitochondria are shown as negative, while those flowing out are positive. Membrane capacitance transients *observed* upon application of voltage steps were removed from current traces.

Typically, pipettes were filled with one of the following three solutions (*Garg and Kirichok, 2019*) (tonicity was adjusted to ~350 mmol/kg with sucrose).

*Solution A* was used to measure $Ca^{2+}$ currents and contained: 110 mM Na-gluconate, 40 mM HEPES, 10 mM EGTA and 2 mM $MgCl_2$ (pH 7.0 with NaOH).

*Solution B* was used to measure $Na^+$ or $Mn^{2+}$ currents and contained: 110 Na-gluconate, 40 HEPES, 1 EGTA, 5 EDTA, and 2 mM NaCl (pH 7.0 with Tris base).

*Solution C* was used to measure outward $Ca^{2+}$ currents (the MCU rectification experiments) and contained: 130 mM tetramethylammonium hydroxide (TMA), 100 mM HEPES and 2 mM $CaCl_2$ (pH 7.0 with D-gluconic acid).

To measure whole-mitoplast $Ca^{2+}$ currents, the bath solution was formulated to contain only 150 mM HEPES (pH 7.0 with Tris base, tonicity ~300 mmol/kg with sucrose) and different dilutions of $CaCl_2$ from a 1 M stock (Sigma) (*Kirichok et al., 2004*). The control solution contained: 150 mM HEPES, 80 mM sucrose and 1 mM EGTA (pH 7.0 with Tris base, tonicity ~300 mmol/kg with sucrose). The bath solution used for measuring $Na^+$ current contained: 110 mM Na-gluconate, 40 mM HEPES, 1 mM EGTA and 5 mM EDTA (pH 7.0 with Tris base, tonicity ~300 mmol/kg with sucrose). The bath solution for measuring inhibition of $Na^+$ current by cytosolic $Ca^{2+}$ contained: 110 mM Na-gluconate, 40 mM HEPES, and 10 mM EDTA (pH 8.0 with Tris, tonicity ~380 mmol/kg with sucrose) and varying amounts of $CaCl_2$ were added to the bath solution to achieve the free $[Ca^{2+}]$ calculated using the MaxChelator program (C. Patton, Stanford University).

A rapid exchange of $[Ca^{2+}]_i$ from virtual zero (control solution) to 1 mM was achieved using a commercially available fast solution exchange system (Warner Instruments, SF-77B perfusion fast step system). It was interfaced with our pClamp acquisition software in order to precisely time the steps during solution change. The timing (τ ~0.4 ms) for solution exchange was judged by the current changes because of a junction potential difference using solutions with different ionic strengths.

Currents were recorded using an Axopatch 200B amplifier (Molecular Devices). Data acquisition and analyses were performed using PClamp 10 (Molecular Devices) and Origin 9.6 (OriginLab). All data were acquired at 10 kHz and filtered at 1 kHz.

## Single-channel recordings and analysis

All single-channel data were acquired from inside–out patches excised from isolated mitoplasts (**Kirichok et al., 2004**). For $Ca^{2+}$ single channel ($i_{Ca}$) recordings, patches were excised in a bath solution containing 150 mM KCl, 10 mM HEPES and 1 mM EGTA, pH 7.2 (adjusted with KOH). Recordings were performed under symmetrical conditions (the same bath and pipette solutions): 105 mM $CaCl_2$ and 40 mM HEPES, pH 7.0 with Tris base. Signals were sampled at 50 kHz and low-pass filtered at 1 kHz. Fire-polished, borosilicate pipettes (Sutter QF-150–75) coated with Silguard (Dow Corning Corp., Midland, MI) and having a tip resistance of 50–70 MΩ were used for low noise recordings.

For $Na^+$ single channel ($i_{Na}$) recordings, patches were excised in a bath solution containing 150 mM Na-gluconate, 10 mM HEPES, 1 mM EGTA and 1 mM $MgCl_2$, pH 7.2 (adjusted with NaOH). Pipette solution contained 150 mM Na-gluconate, 10 mM HEPES, 1 mM EGTA, 1 mM EDTA, and 2 mM NaCl, pH 7.2 (adjusted with NaOH). Signals were sampled at 50 kHz and low-pass filtered at 1 kHz.

To characterize the single-channel conductance and subconductance levels and their occupancy probabilities, we used the MLab version of the QuB software, freely available from the Milescu lab at: https://milesculabs.biology.missouri.edu/QuB_Downloads.html. The data were first resampled at 2.5 kHz and then were idealized with the Baum-Welch and Viterbi algorithms, as implemented in QuB, which classify each point in the data to a conductance level and produce estimates of current amplitudes and occupancy probabilities. The time-averaged single-channel current can be calculated as the product between occupancy probability and current amplitude, summated over all conductance levels (main open state and substates).

## Time-lapse $Ca^{2+}$ imaging in intact cells

For imaging experiments, MEFs were plated on collagen type-I-coated glass-bottom 35 mm dishes (P35G-1.5–14 C, Matek), 48–72 hr before imaging. Cells were imaged at the interval of 3 s on a Nikon Ti-E microscope using a 40× objective (NA 1.30, oil, CFI Plan Fluor, Nikon), Lambda 421 LED light source (Sutter) and ORCA Flash 4.0 CMOS camera (Hamamatsu Photonics) at room temperature (25°C). The following excitation/emission filter settings were used: 340±13/525±25 nm and 389±19/510±40 nm for cytosolic $Ca^{2+}$ imaging using fura-2 ($K_d$=224 nM) and 480±40/525±15 nm for mitochondrially targeted *cepia2* (CEPIA2mt, $K_d$=160 nM (**Suzuki et al., 2014**), cloned into a lentiviral vector). Cells were loaded with 3 µM fura-2 AM (Life Tech., USA) in DMEM/FBS at room temperature for 30 min. After three washes with physiological salt solution (PSS) containing (in mM) 150 NaCl, 4 KCl, 2 $CaCl_2$, 1 $MgCl_2$, 5.6 glucose, and 25 HEPES (pH 7.4), each dish was placed on the stage for imaging. Imaging was performed in PSS within 1 hr of dye staining. Baseline fluorescence was taken for 1–2 min after which thapsigargin (Tg) (final [Tg] = 300 nM) was added while imaging was continued for another 10–15 min.

### Fura-2 calibration

Baseline measurements were taken, and cells were incubated in PSS (No $CaCl_2$) containing 3 mM EGTA, 1 µM ionomycin and 1 µM Tg for 5–10 min. After 2–3 washes with PSS (No $CaCl_2$) containing 0.3 mM EGTA, cells were imaged for 5 min (average of last 10 frames was used for calculation) to obtain the $R_{min}$ and $F_{380max}$ values. Finally, PSS containing 10 mM $CaCl_2$ (no EGTA), 1 µM ionomycin and 1 µM Tg was added and cells were imaged for 10 min. After the signal reached saturation (~3 min), the average value from 10 frames was used to calculate $R_{max}$ and $F_{380min}$ values. Using these obtained values, the fura-2 ratio was calibrated by the following equation (**Grynkiewicz et al., 1985**):

$$[\mathrm{Ca}^{2+}]_{free} = \mathrm{K_d} * \left( \frac{[\mathrm{R} - \mathrm{R_{min}}]}{[\mathrm{R_{max}}]} \right) * (\mathrm{F}_{380max} / \mathrm{F}_{380min})$$

All image analyses were done with ImageJ (NIH). Briefly, mitochondrial and cytosolic regions were manually determined for each cell. The average fluorescence intensity in the regions was measured and the background intensity was subtracted. For analysis of the *cepia2* signal, we normalized the fluorescence intensity by the baseline fluorescence. For analysis of the fura-2 signal, we calculated the fluorescence ratio ($F_{340}/F_{380}$ for fura-2).

The time point for increase in mitochondrial [Ca$^{2+}$] (upstroke) was detected using a script written in Python and manually checked afterwards. Briefly, the fluorescence signal was smoothed by applying a second-order zero phase digital Butterworth filter with an optimal cutoff frequency as previously described (*Winter, 2009*). From the smoothed signal, the upstroke frame was defined as the earliest point between the baseline and signal peak that was greater than 80% of the maximal time derivative. The time-point for change in mitochondrial signal was time-matched with the fura-2 reading to determine the threshold [Ca$^{2+}$]$_i$.

## Measurements of mitochondrial Ca$^{2+}$ influx in isolated mitochondria

Briefly, mitochondria were isolated from MEF cells using differential centrifugation as described above. The mitochondrial pellet was resuspended in resuspension buffer (RB) supplemented with 2 mM EGTA and 2 µM of Fura-2-acetoxymethyl ester (Fura-2 AM) and kept at room temperature for 10 min to allow loading of Fura-2 into the mitochondrial matrix. The RB buffer contained: 100 mM KCl, 50 mM MOPS, 1 mM MgCl$_2$. Mitochondria were pelleted at 3200 g, and further incubated on ice for 50 min to allow de-esterification of Fura-2 AM in RB supplemented with 2 mM EGTA. Mitochondria were pelleted at 3200 g and resuspended in RB supplemented with 10 µM EGTA. Mitochondria were further pelleted and resuspended twice in RB supplemented with 40 µM Fluo-4 pentapotassium salt (for measurements carried out below 3 µM [Ca$^{2+}$]$_i$), or RB supplemented with 40 µM EGTA (for measurements carried out above 3 µM [Ca$^{2+}$]$_i$). After final centrifugation step at 3200 g, protein concentration was determined by Lowry assay.

Measurements of mitochondrial Ca$^{2+}$ influx were carried out using a BMG LABTECH CLARIOstar plate reader as described before (*Wescott et al., 2019*). Experiments were carried out with mitochondria (0.5 mg/ml) in an uptake assay buffer (uAB) that contained: 130 mM KCl, 20 mM HEPES, 1 mM MgCl$_2$, 1 mM K$_2$HPO$_4$, pH 7.2 (with KOH), supplemented with energetic substrates (glutamate, malate and succinate, each 5 mM), and 1 µM TMRM. The uAB and all the stock solutions were made with analytical-grade deionized water (OmniSolv LC–MS, Sigma Aldrich) and contained less than 50 nM of residual [Ca$^{2+}$] (measured daily). After 3 min of incubation with substrates, assays were initiated by injection of 100 µl of Ca$^{2+}$ stock to bring the final volume to 200 µl. TMRM (ex: 546 ± 4 nm and 573 ± 5 nm, em: 619 ± 15 nm) and Fura-2 (ex: 335 ± 6 nm and 380 ± 6 nm, em: 490 ± 15 nm) fluorescence were measured, along with Fluo-4 or Fluo4-FF (ex: 485 nm, em: 520–542 nm) within the same well for 35 s. To measure MCU Ca$^{2+}$ flux ($J_{MCU}$), two protocols were used. Protocol 1 ([Ca$^{2+}$] range $\leq$ 3 µM): here, Fluo-4 (3 µM) is the single significant buffer of extra-mitochondrial Ca$^{2+}$ (i.e., [Ca$^{2+}$]$_i$). Protocol 2 ([Ca$^{2+}$] range from 4 µM to 25 µM): here mitochondria were suspended in 40 µM EGTA and 1 µM Fluo4-FF was used. Total Ca$^{2+}$ influx ($J$) is taken as the first derivative of the linear fit to the measured total extramitochondrial [Ca$^{2+}$] over the first 20 s of each experiment. The total Ca$^{2+}$ conductance of the IMM ($G$) was obtained from the simultaneous measurements of $J$, [Ca$^{2+}$]$_i$, [Ca$^{2+}$]$_m$ and $\Delta\Psi_m$ according to the typical Hodgkin–Huxley model (*Wescott et al., 2019*):

$I = G$ ($\Delta\Psi_m - E_{Ca}^{2+}$) where $E_{Ca}^{2+}$ is the Nernst reversal potential for Ca$^{2+}$ obtained from simultaneously measured [Ca$^{2+}$]$_i$, and [Ca$^{2+}$]$_m$. Measured $J$ was converted to $I$ using the Faraday constant (*Wescott et al., 2019*).

## Measurements of mitochondrial Na$^+$ influx in isolated mitochondria

Membrane potentials in intact mitochondria were evaluated with TMRE using previously described method (*Scaduto and Grotyohann, 1999*). Mitochondria isolated from mouse liver or MEF cells were suspended in ice-cold initial medium. These mitochondria were mixed in 50–100 µl of an uptake assay buffer (liver: 150 mM NaCl, 10 mM HEPES, 1 mM EGTA, 2 mM glutamate, 2 mM malate, and 2 mM succinate, pH 7.2 with Trizma base; MEF: 30 mM NaCl, 120 mM TrisCl, 10 mM HEPES, 1 mM EGTA, 1 µM MgCl$_2$, 5 mM glutamate, 5 mM malate, and 5 mM succinate, pH 7.2 with Trizma base) with 200 nM TMRE. Mitochondrial concentration in the assay buffer was 0.25 mg/ml. TMRE fluorescence were measured at 550/570 nm and 570/589 nm (excitation/emission, 9 nm band width) with using a Biotek Synergy H4 plate reader, and the fluorescence ratio between two fluorescence was calculated. After 5 min incubation of mitochondria in the assay medium, assays were initiated by injection of 0.5–1 µl of EDTA (5 mM final), RuR (1–3 µM final) or FCCP (1 µM final), and the ratio change within 5 min after the injection was evaluated. The ratio change induced by EDTA and/ or RuR was normalized with that by FCCP.

## Co-immunoprecipitation

Mitochondria or mitoplasts were isolated from MEFs deficient in the MCU subunit but stably expressing Flag-tagged MCU. Mitochondrial fraction from wild type cells (without MCU-FLAG) was used as negative control. Isolated mitoplasts (but not mitochondria) were incubated in 750 mM KCl for 30 min before solubilization. Briefly, 300 µg of protein lysate was solubilized with 500 µl of lysis buffer (50 mM HEPES pH 7.4, 150 mM NaCl, 1 mM EGTA, 0.2% DDM and Halt protease inhibitor cocktail [Thermo Fisher]) for 30 min at 4°C. Lysates were cleared by spinning at 20,000× g for 10 min at 4°C. Cleared lysates were incubated with anti-Flag M2 affinity gel (Sigma A2220) for 2 hr at 4°C. Immunoprecipitates were washed with 1 ml of lysis buffer three times and boiled in 20 µl of Laemmli buffer (without β-mercaptoethanol). One-third of the immunoprecipitate was loaded onto a 4–20% gradient SDS-PAGE gel for detection of the indicated proteins by Western blotting. Flow-through fraction was also collected and analyzed in the same gel.

## Immunoblots

For western blot analysis, MEFs or isolated mitochondria/mitoplasts were lysed in radioimmunoprecipitation assay (RIPA) buffer (1% IGEPAL, 0.1% sodium dodecyl sulfate, 0.5% sodium deoxycholate, 150 mM NaCl, 1 mM EDTA, 50 mM Tris-HCl (pH 7.4) and a cocktail of proteases inhibitors). Lysates were resolved by SDS-PAGE; transferred to PVDF membrane (Millipore); and probed with anti-MCU (Sigma, HPA016480, 1:2000), anti-EMRE (Santa Cruz, sc-86337, 1:200), anti-HSP60 (Santa Cruz, sc-1052, 1:3000), anti-VDAC (Abcam, ab15895, 1:2000), anti-MICU1 (Cell Signaling Technology, 12524S, 1:2000), anti-MICU2 (Bethyl, A300-BL19212, 1:500), anti-MICU3 (Sigma, HPA024779, 1:1000), and anti-TOM20 (Santa Cruz, sc-11415, 1:2000). Anti-MICU1 antibody produced a non-specific band near its monomeric molecular weight (~50 kDa), so samples were prepared in Laemmli buffer without β-mercaptoethanol to detect MICU1 homo- or heterodimers (~100 kDa).

## Statistical analysis

Data are presented as mean ± standard error of the mean (SEM), as specified in the Figure legend. Statistical analysis was completed in Excel or Origin 9.6. All experiments were performed in triplicate or more. Statistical significance at an exact p-value was determined with the methods as indicated in the corresponding Figure legends.

# Acknowledgements

We thank Drs. Katsuyoshi Mihara (Kyushu University, Japan) and David C. Chan (Caltech, USA) for sending us DRP1-KO MEFs, and Dr. Toren Finkel (University of Pittsburgh, USA) for sending the MEFs with intact DRP1 (WT and MICU1-KO MEFs) and the MCU-KO mice. We thank the Nikon Microscopy Core (DeLaine Larsen, Kari Herrington) and Lab for Cell Analysis (Sarah Elms) at UCSF for help with use of microscopy and FACS equipment. We thank all members of the YK lab for helpful discussions. This work was supported by American Heart Association Scientist Development Grant 17SDG33660926 (VG) and NIH grant 5R01GM107710 (YK) and R35GM136415 (YK).

# Additional information

## Funding

| Funder | Grant reference number | Author |
| --- | --- | --- |
| National Institutes of Health | R01GM134536 | Yuriy Kirichok |
| National Institutes of Health | R35GM136415 | Yuriy Kirichok |
| American Heart Association | 17SDG33660926 | Vivek Garg |

The funders had no role in study design, data collection and interpretation, or the decision to submit the work for publication.

## Author contributions
Vivek Garg, Conceptualization, Resources, Software, Formal analysis, Supervision, Funding acquisition, Validation, Investigation, Visualization, Methodology, Writing - original draft, Project administration, Writing - review and editing; Junji Suzuki, Formal analysis, Investigation, Visualization, Methodology; Ishan Paranjpe, Software, Investigation; Tiffany Unsulangi, Investigation; Liron Boyman, Investigation, Methodology, Writing - review and editing; Lorin S Milescu, Software- Formal analysis; W Jonathan Lederer, Resources, Supervision, Methodology, Writing - review and editing; Yuriy Kirichok, Conceptualization, Resources, Data curation, Supervision, Funding acquisition, Methodology, Writing - original draft, Project administration, Writing - review and editing

## Author ORCIDs
Vivek Garg (iD) https://orcid.org/0000-0002-6940-5415
Yuriy Kirichok (iD) https://orcid.org/0000-0001-7155-843X

## Ethics
Animal experimentation: All animal experiments were performed according to procedures approved by the UCSF Institutional Animal Care and Use Committee (approval # AN183460-02A) and adhered to NIH standards.

## Decision letter and Author response
Decision letter https://doi.org/10.7554/eLife.69312.sa1
Author response https://doi.org/10.7554/eLife.69312.sa2

## Additional files

### Supplementary files
• Supplementary file 1. MICU1 effect on $MCU_{cx}$. (a) MICU1 effect on $MCU_{cx}$ as determined by previous electrophysiological experiments. (b) MICU1 effect on $MCU_{cx}$ as determined by previous $Ca^{2+}$ imaging experiments.

• Transparent reporting form

### Data availability
Due to the size of the dataset, raw electrophysiology traces are available on request to the corresponding author. All information has been extracted from the raw electrophysiological traces and is available to download as source data files. All the codes or software used in analyzing the data and their sources are listed in the Key Resources Table.

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
