## [Decision Letter]

**Acceptance summary:**

This paper examines the roles and mechanisms of how subunits of the mitochondrial calcium uniporter complex (MCU_cx_) regulate calcium uptake by mitochondria, a process that serves to match the rate of ATP generation to cellular metabolic needs. Based on direct electrophysiological recordings of MCU_cx_, the authors find that the MICU1 subunit potentiates channel activity in a calcium-dependent manner but does not block the channel at low calcium levels, challenging current models of MCU regulation. This work will be of significant interest to biophysicists and cell biologists interested in mitochondrial biology, bioenergetics, and ion channel and calcium signaling mechanisms.

**Decision letter after peer review:**

Thank you for submitting your article "The Mechanism of MICU-Dependent Gating of the Mitochondrial Ca^2+^ Uniporter" for consideration by *eLife*. Your article has been reviewed by 3 peer reviewers, one of whom is a member of our Board of Reviewing Editors, and the evaluation has been overseen by Kenton Swartz as the Senior Editor. The following individual involved in review of your submission has agreed to reveal their identity: Youxing Jiang (Reviewer #3).

Essential revisions:

1. A major conclusion of the paper is that MICU1 does not block the MCU pore at low levels of Ca^2+^. This is supported by Na^+^ flux through MCU under conditions of 0 Ca^2+^ and Mg^2+^ to allow Na^+^ permeation (Figure 2C-I and Figure 5). However, Ca^2+^ uptake at low levels of Ca^2+^ is apparently higher in MICU1^-/-^ cells compared to WT cells (Figure 1F-1I), consistent with the idea that MICU1 does inhibit MCU_cx_ at resting intracellular Ca^2+^. A weakness of the manuscript is that no mechanism is investigated or proposed for this effect of MICU1. Rather, this result is explained away by vague arguments that MICU1 KO might change Ca^2+^ efflux, buffering, pH, or OMM permeability (lines 365-7). A more serious attempt should be made to understand the basis of this effect; new experimental evidence or a more detailed mechanism based on published work should be presented to explain how MICU1 could prevent the rise of free mitochondrial Ca^2+^ in the presence of low rate of influx. A potential plausible mechanism is that one particular EF hand is responsible for this low Ca^2+^ blocking effect while another mediates the high Ca^2+^- dependent potentiation of MCU_cx_ open probability.

Also, the authors imply that structural evidence for MICU1 blocking the pore was only obtained for the monomeric complex, and that the more natural dimeric complex did not show block (l. 408-418). This is not strictly true (see Wang et al. 2020b). Also, while it is possible that low ionic strength could lead to artifactual plugging in the structures (l. 421-3), to be fair it should also be noted that the plug model is consistent with increased Ca^2+^ uptake by mutations predicted to disrupt electrostatic interactions in the plug structure (Fan et al. 2020).

2. Strong evidence in mitoplasts and intact mitochondria shows that MICU1 does not affect Na^+^ flux through MCU in the absence of Ca^2+^ and Mg^2+^, arguing against its function as a pore blocker (Figure 2C-I and Figure 5). However, it is possible that the removal of all divalent cations to permit Na^+^ permeation prevents MICU1 from plugging the pore (the use of EDTA to remove Mg^2+^ should be noted in the text, rather than stating only that Ca^2+^ was removed). Given the number of previous functional and structural studies that support a blocking role for MICU1, it is important to rule out a possible dependence on Mg^2+^. If Na^+^ can permeate in the presence of Mg^2+^, then the Na^+^ current measurements or mitochondrial depolarization experiments could be repeated with Mg^2+^ present. If this is not possible, then Mg^2+^ could be removed in the Ca^2+^ uptake experiments (Figure 1F-I) to see whether 0 Mg^2+^ phenocopies the MICU1 KO effect on uptake at low Ca^2+^.

3. Interpretation of mut-EF-MICU2 data (Figure 3C). Authors indicate dominant negative effect of mut-EF-MICU2 is due to displacement of MICU1 from MICU1 homodimers. Presumably, WT MICU2 also displaces the MICU1 from MICU1 homodimers. The results provided suggest that MICU1 homodimers are functionally equivalent to MICU1/MICU2 heterodimers. It is shown that MICU2 knockout increases MICU1 homodimers. What happens with MICU2 expression in MICU1 knockout? Is the ~100 kDa band in the MICU2 blot (Figure S1H) actually representative of MICU1/MICU2 heterodimer?

4. The experiments on kinetics of the MICU1 potentiation (l. 227-230, Figure 3E) need a clearer interpretation.

a) The kinetics of Ca^2+^ current on addition of Ca^2+^ are not affected by MICU1. Rather than describing potentiation as "instantaneous" (it must have a finite response time after all, which simply cannot be detected with the perfusion method), it may be more meaningful to describe the result in a physiological context; e.g., potentiation is fast enough that MICU1 and mitochondria will faithfully track changes in cytosolic [Ca^2+^] which are typically quite slow (order of tens-hundreds of msec).

b) The interpretation of Figure 3E depends on the solution switching speed not being rate-limiting relative to the current response. The SF-77B perfusion system is described as piezoelectric (l. 748-753), but to my knowledge this is driven by a stepper motor, and it is not clear how you could get such a fast switching time (tau = 0.4 ms) from this system. Thus, it might be rate-limiting for the response. It may be helpful to show the time course of solution exchange superimposed on the traces in Figure 3E. If the solution exchange is rate-limiting for these responses, then one cannot draw any conclusions about the speed of MICU1 potentiation response.

5. The lack of outward flux through MCU supports the conclusion that it is a one-way portal (l. 241, Figure 3G), but alternatives should also be considered. Is it possible that outward flux through MCU requires prolonged depolarization such as would occur in vivo? This could be easily tested in whole-mitoplast recordings using a holding potential that mimics the depolarized voltages used in the previous cited studies.

6. The conclusions that MCU is not regulated by matrix Ca^2+^ and the MICU does not plug the channel assume that whole-mitoplast recordings preserve the normal regulation of the channel. It is likely that diffusible molecules are lost by dialysis into the recording pipette. The authors should discuss whether and why they think this is not a problem.

---

## [Author Response]

Essential revisions:1. A major conclusion of the paper is that MICU1 does not block the MCU pore at low levels of Ca^2+^. This is supported by Na^+^ flux through MCU under conditions of 0 Ca^2+^ and Mg^2+^ to allow Na^+^ permeation (Figure 2C-I and Figure 5). However, Ca^2+^ uptake at low levels of Ca^2+^ is apparently higher in MICU1^-/-^ cells compared to WT cells (Figure 1F-1I), consistent with the idea that MICU1 does inhibit MCU_cx_ at resting intracellular Ca^2+^. A weakness of the manuscript is that no mechanism is investigated or proposed for this effect of MICU1. Rather, this result is explained away by vague arguments that MICU1 KO might change Ca^2+^ efflux, buffering, pH, or OMM permeability (lines 365-7). A more serious attempt should be made to understand the basis of this effect; new experimental evidence or a more detailed mechanism based on published work should be presented to explain how MICU1 could prevent the rise of free mitochondrial Ca^2+^ in the presence of low rate of influx. A potential plausible mechanism is that one particular EF hand is responsible for this low Ca^2+^ blocking effect while another mediates the high Ca^2+^- dependent potentiation of MCU_cx_ open probability.Also, the authors imply that structural evidence for MICU1 blocking the pore was only obtained for the monomeric complex, and that the more natural dimeric complex did not show block (l. 408-418). This is not strictly true (see Wang et al. 2020b). Also, while it is possible that low ionic strength could lead to artifactual plugging in the structures (l. 421-3), to be fair it should also be noted that the plug model is consistent with increased Ca^2+^ uptake by mutations predicted to disrupt electrostatic interactions in the plug structure (Fan et al. 2020).

The reviewers raise an important question, why there is an apparent discrepancy between the findings of two methodological approaches. Why at low levels of cytosolic [Ca^2+^], measurements of mitochondrial Ca^2+^ uptake do not detect a Ca^2+^ influx, while electrophysiological measurements find no evidence for conduction occlusion of MCU_cx_ by MICUs. Our short answer is that measurements of mitochondrial Ca^2+^ uptake do not exclusively measure conductance by the MCU_cx_ while the electrophysiological recording presented here do. We examined MCU currents, a direct measurement, and provide new evidence that could partially explain why the net uptake (uptake minus efflux) is low. In newly added experiments (Figure 7) we now show that the conduction pathway of MCU_cx_ is not plugged by MICU1-- even at physiological levels of [Mg^2+^]. Instead, Mg^2+^ strongly blocks the selectivity filter of the pore. In addition, the unitary conductance of an open MCU_cx_ channel at 100 nM [Ca^2+^] is extremely low. The MCU_cx_ flux is thus limited not only because of the low concentration of the conducting ion but also because of Mg^2+^ block. This allows Ca^2+^ efflux machinery to effectively compete with MCU_cx_^-^mediated Ca^2+^ uptake at low cytosolic [Ca^2+^] to reduce net Ca^2+^ accumulation to nearly zero.

“Why does knocking out MICU1 tends to increase the net mitochondrial Ca^2+^ uptake at low levels of [Ca^2+^]?” This is a question that our study does not directly examine. However, we can speculate -- based on recent other studies (see Gottschalk et al., 2019; Tomar et al., 2019; Tufi et al., 2019) -- that knocking out MICU1 affects multiple mitochondrial systems and not only the MCU_cx_. Here, we investigated the specific role that MICU1 play as part of the channel complex and do this by directly examining the MCU_cx_ current. We see no evidence that supports the hypothesis that MICU1 acts as a plug of the pore. Instead, our work shows that at elevated levels of [Ca^2+^], Ca^2+^ binding to the EF hands of the MICU1 works to double the open probability of MCU_cx_. The binding of Ca^2+^ to the EF hands of MICUs, or any consequential effects of this binding occur at levels of [Ca^2+^] that are higher than 100 nM. Indeed, the available titration data demonstrates that the affinity of the MICUs for Ca^2+^ is not high enough for EF hands occupancy to occur at 100 nM. Even for MICU1, which has the highest affinity for Ca^2+^ of all MICUs, no significant binding occurs at 100 nM (resting cytosolic Ca^2+^), and complete saturation of binding would only occur around 3-6 μM (Kamer et al., 2017) (Figures 1E, 1G, 2F, 2H, 2J and 3H). Thus, the quantitative evidence also suggests that Ca^2+^ occupancy of the EF hands at resting cytosolic Ca^2+^ is too low to explain any profound occlusion.

We agree with the reviewers comment that Wang et al., 2020b showed the possibility of the occlusion in the native dimeric form of the MCU_cx_. However, the occluded dimeric complex represented only ~10% of the total number of analyzed particles in the absence of Ca^2+^. The low prevalence of the dimeric complexes can be purely due to the experimental limitations, but, regardless, a more thorough analysis of occlusion in this native form of MCU_cx_ is needed. Also, such structural analysis should be performed in the presence of physiological concentrations of Mg^2+^, while so far Mg^2+^ was absent in all structural studies of the MCU_cx_ holocomplex. We have rewritten the section discussing MCU_cx_ structures to reflect these changes.

Although the occlusion model appears to be consistent with some of the mutations that disrupt electrostatic interactions in the plug structure, all such studies were performed using indirect assessment of MCU function. Such mutations can cause MICUs loss-of-function effects (that is not necessarily loss of occlusion) and could lead to activation of compensatory mechanisms that create an appearance of “the loss of the threshold”, similar to that observed in MICU1 knockout. To interpret the structures correctly, reliable direct functional data is much preferred, as it has always has been the case in the ion channel field.

2. Strong evidence in mitoplasts and intact mitochondria shows that MICU1 does not affect Na^+^ flux through MCU in the absence of Ca^2+^ and Mg^2+^, arguing against its function as a pore blocker (Figure 2C-I and Figure 5). However, it is possible that the removal of all divalent cations to permit Na^+^ permeation prevents MICU1 from plugging the pore (the use of EDTA to remove Mg^2+^ should be noted in the text, rather than stating only that Ca^2+^ was removed). Given the number of previous functional and structural studies that support a blocking role for MICU1, it is important to rule out a possible dependence on Mg^2+^. If Na^+^ can permeate in the presence of Mg^2+^, then the Na^+^ current measurements or mitochondrial depolarization experiments could be repeated with Mg^2+^ present. If this is not possible, then Mg^2+^ could be removed in the Ca^2+^ uptake experiments (Figure 1F-I) to see whether 0 Mg^2+^ phenocopies the MICU1 KO effect on uptake at low Ca^2+^.

We have added a set of new electrophysiological experiments to address the effect of Mg^2+^ on Ca^2+^ currents to answer this question. Please see the new Figure 7. These experiments examine how Mg^2+^ affects the Ca^2+^ conduction through MCU_cx_. There are two main conclusions. First, Mg^2+^ interacts with the selectivity filter within the pore of MCU_cx_ to occlude Ca^2+^ permeation. This effect is completely MICU-independent. Second, the Ca^2+^-dependent potentiating effect of MICUs on I_Ca_ does not depend on Mg^2+^.

We would also like to reemphasize that in this study we did not center our investigation on the net mitochondrial Ca^2+^ uptake, but focus specifically on MCU_cx_ activity. The increase in net mitochondrial Ca^2+^ uptake in MICU1-KO vs WT was observed both in the presence (Csordas et al., 2013) or absence of Mg^2+^ (Mallilankaraman et al., 2012). Thus, the putative occlusion of the MCU pore by MICUs, if it exists, would be a Mg^2+^-independent phenomenon.

With these new results included in the manuscript, we revised the results and the Discussion sections. We now highlight the impact that physiological Mg^2+^ block has, in limiting MCU_cx_ flux at low Ca^2+^. We also emphasize the importance of reevaluating the structure of MCU holocomplex in the Mg^2+^ bound conformation. We thank the reviewers for prompting this addition.

Per reviewers’ request, we now clearly indicate in the text that the use of EDTA removes not only Ca^2+^ but also Mg^2+^.

3. Interpretation of mut-EF-MICU2 data (Figure 3C). Authors indicate dominant negative effect of mut-EF-MICU2 is due to displacement of MICU1 from MICU1 homodimers. Presumably, WT MICU2 also displaces the MICU1 from MICU1 homodimers. The results provided suggest that MICU1 homodimers are functionally equivalent to MICU1/MICU2 heterodimers. It is shown that MICU2 knockout increases MICU1 homodimers. What happens with MICU2 expression in MICU1 knockout? Is the ~100 kDa band in the MICU2 blot (Figure S1H) actually representative of MICU1/MICU2 heterodimer?

MICU1 is the primary subunit responsible for tethering of other MICUs (MICU2 and MICU3) to the MCU/EMRE pore. In the absence of MICU1, MICU2 cannot bind to the pore. This has been shown in many previous studies both biochemically and structurally. MICU2 expression remains unchanged or a significant amount gets degraded in the absence of MICU1 (Kamer and Mootha, 2014; Patron et al., 2014; Payne et al., 2017).

The 100 kDa band in the Figure 1—figure supplement 1H (previously, Figure S1H) represents MICU2 in the MICU1/MICU2 heterodimer. Although the protein samples in the Figure 1—figure supplement 1H were prepared under the reducing conditions (i.e. with β-mercaptoethanol), we observed bands near the molecular weights corresponding to both the monomeric and heterodimeric forms. For detection of MICU1, the samples were prepared under non-reducing conditions, so the band was always observed at the molecular weight corresponding to the dimeric form. This is mentioned in the Methods section and in the corresponding legends.

4. The experiments on kinetics of the MICU1 potentiation (l. 227-230, Figure 3E) need a clearer interpretation.a) The kinetics of Ca^2+^ current on addition of Ca^2+^ are not affected by MICU1. Rather than describing potentiation as "instantaneous" (it must have a finite response time after all, which simply cannot be detected with the perfusion method), it may be more meaningful to describe the result in a physiological context; e.g., potentiation is fast enough that MICU1 and mitochondria will faithfully track changes in cytosolic [Ca^2+^] which are typically quite slow (order of tens-hundreds of msec).b) The interpretation of Figure 3E depends on the solution switching speed not being rate-limiting relative to the current response. The SF-77B perfusion system is described as piezoelectric (l. 748-753), but to my knowledge this is driven by a stepper motor, and it is not clear how you could get such a fast switching time (tau = 0.4 ms) from this system. Thus, it might be rate-limiting for the response. It may be helpful to show the time course of solution exchange superimposed on the traces in Figure 3E. If the solution exchange is rate-limiting for these responses, then one cannot draw any conclusions about the speed of MICU1 potentiation response.

(a) Thank you for the suggestion. We have modified the manuscript to describe the results in a physiological context as was suggested by the reviewers.

(b) Thank you for pointing it out. We acknowledge the error in describing the system. SF77B is a stepper motor. We now show the time-course for the solution exchange in the supplementary Figure 3—figure supplement 2. The time constant of solution exchange was ~0.4 ms. The solution exchange was performed using a thinly-pulled glass theta tubing (double-barreled tube). The mitoplasts (which are much smaller than a cell) were very close to the perfusion. Because mitoplasts are comparable in size to the tip of the pipette, the rate of solution exchange in actual experiments will be similar.

5. The lack of outward flux through MCU supports the conclusion that it is a one-way portal (l. 241, Figure 3G), but alternatives should also be considered. Is it possible that outward flux through MCU requires prolonged depolarization such as would occur in vivo? This could be easily tested in whole-mitoplast recordings using a holding potential that mimics the depolarized voltages used in the previous cited studies.

MCU is a highly inwardly rectifying channel. Previous single and multichannel inside-out recordings (with 105 mM matrix Ca^2+^) at positive potentials (> 0 mV) show rapid flickering of the outward unitary currents as compared to unoccluded square openings in the inward directions (Figure 4a, 4d, and S2a) (Kirichok et al., 2004). However, these recordings demonstrate that efflux of Ca^2+^ via MCU_cx_ is possible, at least at 105 mM Ca^2+^ on the matrix side of the IMM. Importantly, these single-channel recording also showed that the Ca^2+^ efflux via MCU_cx_ does not change with time during prolonged depolarizations. In the whole-mitoplasts experiments presented in the current manuscript, we could not use 105 mM Ca^2+^ on the matrix side of the IMM (pipette solution), and even 2 mM Ca^2+^ is barely tolerated, as the membrane integrity is compromised. We cannot exclude a possibility that at 2 mM matrix Ca^2+^ the outward current via MCU_cx_ is so small that the patch-clamp electrophysiology is unable to resolve it. So, as reviewer implied, we should not claim that Ca^2+^ efflux via MCU_cx_ is impossible. We now rewrite this part of the manuscript to reflect this.

That said, the data as presented still show a dramatic inward rectification of MCU, and that MICUs are not the cause of this rectification. Again, we rewrote this section of the manuscript to modify our interpretation of these data.

6. The conclusions that MCU is not regulated by matrix Ca^2+^ and the MICU does not plug the channel assume that whole-mitoplast recordings preserve the normal regulation of the channel. It is likely that diffusible molecules are lost by dialysis into the recording pipette. The authors should discuss whether and why they think this is not a problem.

The manuscript specifically addresses the current model of MICU-dependent occlusion of MCU_cx_, and this model does not require any matrix regulators for the occlusion to happen. Importantly, the cryo-EM structures that suggest the MCU pore is occluded by MICU1 also do not require any diffusible matrix molecules. Using patch clamp methodology, one group (Vais et al., 2016; Vais et al., 2020) suggested that there is a biphasic regulation of MCU_cx_ by matrix Ca^2+^ ions and this regulation is MICU-dependent. We addressed this possibility but saw no regulation of MCU_cx_ activity by matrix Ca^2+^ within the physiological range. Regulation of MICU function by matrix Ca^2+^ appeared plausible at the time when it was still not clear whether MICUs are located on the cytosolic or matrix side of the IMM. Now that the structural data clearly showed MICUs location on the cytosolic face of the IMM, regulation of MICU function by matrix Ca^2+^ is poorly justified.

MCU_cx_ is a large complex and could potentially be regulated by some diffusible matrix molecules. However, these are unlikely to affect MICUs function as MICU proteins are located on the cytosolic face of the IMM.